# Enhancing CLIP Robustness via Cross-Modality Alignment

**Xingyu Zhu[1,2]**   **Beier Zhu[2]**   **Shuo Wang[1†]**   **Kesen Zhao[2]**   **Hanwang Zhang[2]**
[1]University of Science and Technology of China
[2]Nanyang Technological University
xingyuzhu@mail.ustc.edu.cn,   shuowang.edu@gmail.com

## Abstract

Vision-language models (VLMs) such as CLIP demonstrate strong generalization in zero-shot classification but remain highly vulnerable to adversarial perturbations. Existing methods primarily focus on adversarial fine-tuning or prompt optimization, they often overlook the gaps in CLIP's encoded features, which is shown as the text and image features lie far apart from each other. This misalignment is significantly amplified under adversarial perturbations, leading to severe degradation in classification performance. To address this problem, we propose **CrO**ss-moda**L**ity **A**lignment, dubbed **COLA**, an optimal transport-based framework that explicitly addresses adversarial misalignment by restoring both global image-text alignment and local structural consistency in the feature space. (1) COLA first projects adversarial image embeddings onto a subspace spanned by class text features, effectively filtering out non-semantic distortions while preserving discriminative information. (2) It then models images and texts as discrete distributions over multiple augmented views and refines their alignment via OT, with the subspace projection seamlessly integrated into the cost computation. This design ensures stable cross-modal alignment even under adversarial conditions. COLA is training-free and compatible with existing fine-tuned models. Extensive evaluations across 14 zero-shot classification benchmarks demonstrate the effectiveness of COLA, especially with an average improvement of 6.7% on ImageNet and its variants under PGD adversarial attacks, while maintaining high accuracy on clean samples.

## 1   Introduction

Vision-language models (VLMs) [17, 30] like CLIP [45] demonstrate strong generalization ability in zero-shot classification. However, they are highly susceptible to adversarial perturbations, where small but carefully crafted changes to input images can significantly mislead predictions [31, 32, 36]. Such vulnerabilities pose serious risks in critical applications such as medical diagnosis, autonomous driving, and security systems, where robustness and reliability are paramount.

Recent efforts to improve the adversarial robustness of VLMs can be broadly categorized into three directions: adversarial training [4, 62], which fine-tunes models with perturbed samples; prompt tuning [31, 63], which optimizes text input templates to resist attacks, and test-time defenses [21, 2, 59], which modify inputs or predictions on the fly. While these methods offer promising improvements, they suffer from high computational overhead or introduce substantial inference latency. More critically, they overlook a central issue: the misalignment between image and text modalities [70, 16]. This misalignment stems from CLIP's global matching paradigm, where the model is trained to align entire image embeddings with sentence-level textual embeddings. As shown in Figure 1(a), the text

---

†Corresponding author

39th Conference on Neural Information Processing Systems (NeurIPS 2025).

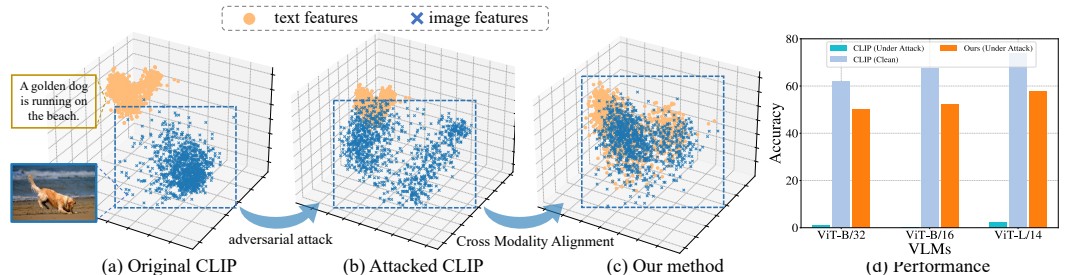

Figure 1: Visualization of image and text feature distributions under different conditions. We plot text and image embeddings via Principal component analysis (PCA) [1] and compare their performance. (a) Adversarial perturbations cause image features to scatter and misalign with text features. (b) Clean image and text features naturally form two distinct clusters as a result of contrastive training. (c) Our method mitigates the misalignment, making adversarial image features closer to the text features. (d) Classification performance across multiple VLMs shows that our method consistently improves robustness against adversarial inputs.

"*A golden dog running on the beach*" offers a fine-grained description that includes the object, its attributes, and the surrounding background. However, the image encoder processes the entire scene as a global representation, without explicitly modeling how these textual elements correspond to specific regions of the image. This results in image and text features being distributed independently in separate regions of the embedding space.

Such misalignment becomes particularly problematic under adversarial attacks [15, 31]. As illustrated in Figure 1(b), even small perturbations can distort the image embedding and severely disrupt global feature alignment, pushing visual representations away from their semantic prototypes. Beyond global shifts, attacks also damage the local structure within the feature space, causing nearby image embeddings to scatter and lose their internal consistency. As shown in Figure 1(d), this dual breakdown of alignment leads to a near-collapse in classification accuracy.

To address this issue, we propose a training-free framework that explicitly addresses adversarial misalignment at both the feature and semantic levels. First, we project adversarial image embeddings onto a text-induced subspace, eliminating non-semantic distortions and restoring feature space alignment. Then, we model images and texts as discrete distributions over multiple augmented views and refine their correspondence through optimal transport (OT) based on the projected features. Subspace projection is directly embedded into the OT cost, and we theoretically guarantee that it does not increase the transport distance. By jointly aligning feature embeddings and semantic distributions, our approach substantially improves the adversarial robustness of CLIP. Figure 1(d) illustrates the accuracy improvement over the attacked CLIP.

We conduct extensive experiments across 14 zero-shot classification benchmarks to evaluate the effectiveness of our method. Results demonstrate that COLA substantially improves adversarial robustness under multiple attack settings, with notable improvements such as an average gain of 6.7% under PGD and 4.8% under CW attacks on ImageNet and its variants, while maintaining high accuracy on clean images. Moreover, COLA can be directly applied to different CLIP fine-tuned models without any retraining, making it practical for real-world deployment.

## 2 Related Work

**Adversarial robustness in VLMs.** Adversarial robustness remains a fundamental challenge, as small, imperceptible perturbations can mislead model predictions [51, 7]. A common defense is adversarial training (AT) [33, 62, 47], which improves robustness but incurs high computational cost [50, 56]. Recent test-time defenses—such as generative purification [37, 61] and optimization-based methods [57, 35]—offer alternatives but often fail under adaptive attacks [12]. Hedge Defense (HD) [57], for example, perturbs inputs by maximizing cross-entropy loss, but requires an adversarially trained model. Meanwhile, several works explore CLIP's [45] robustness, noting its natural tendency to deflect attacks via counteractive perturbations in latent space. To further improve performance, researchers have applied adversarial fine-tuning [36, 54] and prompt tuning with frozen weights [31, 63].

These methods enhance robustness but rely on training. In contrast, our work proposes the first test-time defense for CLIP that is training-free, architecture-free, and efficient at inference. In contrast to prior efforts that rely on adversarial training, prompt tuning, or additional inference-time modules, we introduce a simple yet effective test-time defense for CLIP, which improves adversarial robustness by restoring image-text alignment through subspace projection and distribution-level matching, without requiring any model retraining or architectural changes.

**Optimal transport.** Optimal transport offers a principled way to compare probability distributions by capturing their geometric relationships [41]. With the development of efficient solvers such as the Sinkhorn algorithm [13], OT has been widely applied to tasks including generative modeling [3, 43, 44], domain adaptation [11], and structural alignment [9, 60]. In the vision-language domain, OT has enabled fine-grained alignment of image-text distributions in few-shot learning [27, 67], distribution calibration [22, 68], and prompt learning [8, 52]. Of particular relevance are recent OT-based methods for vision-language modeling [71], which improve zero-shot performance by enhancing alignment between visual and textual modalities. However, these approaches typically rely on training-time optimization or prompt tuning. While prior works focus on training-time alignment or require prompt tuning, our approach differs in that it introduces an efficient test-time OT framework for adversarially perturbed images. Specifically, we use OT to align projected image features with augmented textual prototypes, thereby enhancing robustness without any model fine-tuning.

## 3 Method

We propose a unified OT framework to enhance robust zero-shot classification by simultaneously addressing the modality misalignment caused by adversarial distortions and the mismatch between images and their text descriptions. We further provide theoretical guarantees that our method better preserves semantic similarity and yields larger margins, suggesting improved generalization.

### 3.1 Preliminaries

**Zero-shot classification.** CLIP [45] consists of a vision encoder $\Phi_v(\cdot)$ and a text encoder $\Phi_t(\cdot)$. Given a set of $K$ class names $\{z_y\}_{y=1}^K$ and a hand-crafted template $G$, *e.g.*, "a photo of a $z_y$", the textual feature for class $y$ is computed as $\mathbf{z}_y = \Phi_t(G(z_y))$. The visual feature for a testing samples $x$ is calculated as $\mathbf{x} = \Phi_v(x)$, where both $\mathbf{x}$ and $\mathbf{z}_y$ lie in the same $d$-dimensional embedding space ($\mathbf{x}, \mathbf{z}_y \in \mathbb{R}^d$). CLIP performs classification by comparing the similarity between the visual feature $\mathbf{x}$ and all text prototypes $\{\mathbf{z}_y\}_{y=1}^K$:

$$y = \underset{y \in [K]}{\arg\max} \, \mathbf{z}_y^\top \mathbf{x}. \tag{1}$$

Recent practices [42, 29, 48] replace the hand-crafted prompt $G(z_y)$ with a set of fine-grained class descriptions $\{\tilde{z}_y^m\}_{m=1}^M = \mathsf{LLM}(z_y)$ generated by large language models (LLMs). The corresponding text features $\{\mathbf{z}_y^m\}_{m=1}^M$ for class $j$ are obtained via $\mathbf{z}_y^m = \Phi_t(\tilde{z}_j^m)$, and the average feature $\bar{\mathbf{z}}_y = \frac{1}{M} \sum_{m=1}^M \mathbf{z}_y^m$ is used in place of $\mathbf{z}_y$ in Eq. (1) for classification.

**Adversarial perturbations.** When the attacker has full access of model parameters, it becomes vulnerable to adversarial perturbations $\delta_a$, which are typically generated via methods such as Projected Gradient Descent (PGD) [7]:

$$\delta_a = \arg\max_\delta L(x_i + \delta, y_i), \text{ s.t. } \|\delta\|_p \leq \epsilon_a \tag{2}$$

where $y_i$ is the ground-truth and $L$ is a loss function, typically cross-entropy. $\delta_a$ is constrained by an $\ell_p$-norm budget $\epsilon_a$, making it visually imperceptible yet highly effective at degrading accuracy.

### 3.2 Cross Modality Alignment under a Unified OT Framework

Original CLIP aligns clean images and texts into a unified feature space, but adversarial attacks on the visual modality severely disrupt this alignment. Moreover, visual features often capture background or irrelevant objects that are not reflected in LLM-generated descriptions, introducing

further semantic misalignment. In this work, we propose a unified OT framework to mitigate both types of misalignment by introducing feature space alignment and local semantics alignment.

**Global feature alignment.** Despite the contamination in image features, the subspace spanned by clean textual features serves as a reliable proxy for reconstructing the underlying clean image representations, a design inspired by [65]. Specifically, we arrange all class text embeddings $\{\mathbf{z}_y^m\}_{y,m}$ into a matrix $\mathbf{Z} \in \mathbb{R}^{d \times KM}$ and apply singular value decomposition (SVD) to extract the top-$C$ principal components:

$$\mathbf{Z} = \mathbf{U}\boldsymbol{\Sigma}\mathbf{V}^\top, \quad \mathbf{U}_C = \mathbf{U}_{[:,1:C]}. \tag{3}$$

This defines a subspace $\mathcal{U} = \mathrm{span}(\mathbf{U}_C)$, which captures $C$ dominant directions shared across class embeddings. Since adversarial perturbations distort image features along directions away from $\mathcal{U}$, we project each perturbed image feature $\hat{\mathbf{x}}$ onto $\mathcal{U}$ to achieve alignment:

$$\Pi(\hat{\mathbf{x}}) = \mathbf{U}_C \mathbf{U}_C^\top \hat{\mathbf{x}}. \tag{4}$$

In Sec. 3.3, we show that the projection helps recover the pairwise similarity of clean image features.

**Local structural alignment.** While feature space alignment restores a unified space, projected image features can still misalign due to visual cues like background or irrelevant objects absent from LLM-generated text. To bridge this gap, we perform local semantics alignment for visual and textual representations. Specifically, for each adversarial image $\hat{x}$, we generate $N-1$ augmented views via random cropping, flipping, or resizing, and include the original to form a set $\{\hat{x}^n\}_{n=1}^N$, which are encoded into features $\{\hat{\mathbf{x}}^n\}_{n=1}^N$. Similarly, for each class name $z_y$, we obtain $M$ textual descriptions by prompting LLMs to generate $M-1$ fine-grained variants in addition to the hand-crafted prompt, yielding features $\{\mathbf{z}_y^m\}_{m=1}^M$. We model each image and class as a discrete distribution, rather than a single embedding. For example, for an image $\hat{x}$ and class $y$, we model their distribution as:

$$\mathbb{P}(\mathbf{x}) = \sum_{n=1}^N a^n \delta(\hat{\mathbf{x}}^n - \mathbf{x}), \quad \mathbb{Q}_y(\mathbf{z}) = \sum_{m=1}^M b_y^m \delta(\mathbf{z}_y^m - \mathbf{z}), \tag{5}$$

where $\delta(\cdot)$ denotes the Dirac delta function, and $a^n$, $b_y^m$ are the associated importance weights. To compute $a^n$ for the augmented image feature $\hat{\mathbf{x}}^n$, we assess its entropy with respect to the average class embedding $\bar{\mathbf{z}}_y = \frac{1}{M}\sum_{m=1}^M \mathbf{z}_y^m$. Specifically, we define:

$$a^n = \frac{\exp\left(h(\hat{\mathbf{x}}^n)\right)}{\sum_{n'=1}^N \exp\left(h(\hat{\mathbf{x}}^{n'})\right)}, \quad h(\hat{\mathbf{x}}^n) = -\sum_{y=1}^K p(\bar{\mathbf{z}}_y|\hat{\mathbf{x}}^n) \log p(\bar{\mathbf{z}}_y|\hat{\mathbf{x}}^n). \tag{6}$$

The entropy $h(\hat{\mathbf{x}}^n)$ reflects the prediction confidence: views with lower entropy are assigned higher weights. The importance weights $b_y^m$ for textual features are computed analogously.

**Unified OT framework.** Given the distributions $\mathbb{P}(\mathbf{x})$ and $\mathbb{Q}_y(\mathbf{z})$, the alignment between an adversarial image and each class is measured by the ot distance, which captures the minimal semantic matching cost between image and text features. We seek a transport plan $\mathbf{T}_y \in \mathbb{R}^{N \times M}$ that that moves mass from $\mathbb{P}(\mathbf{x})$ to $\mathbb{Q}_y(\mathbf{z})$, subject to the marginal constraints:

$$d_{\mathrm{OT}}(\mathbb{P}(\mathbf{x}), \mathbb{Q}_y(\mathbf{z}); \mathbf{C}_j) = \min_{\mathbf{T}_y \geq \mathbf{0}} \langle \mathbf{T}_y, \mathbf{C}_y^\Pi \rangle, \quad \text{s.t.} \quad \mathbf{T}_y \mathbf{1}_M = \mathbf{a}, \quad \mathbf{T}_y^\top \mathbf{1}_N = \mathbf{b}_j, \tag{7}$$

where $\mathbf{a} = [a^1, \cdots, a^N]^\top$ and $\mathbf{b}_y = [b_y^1, \cdots, b_y^M]^\top$, and $\mathbf{1}_N$, $\mathbf{1}_M$ are all-ones vectors. $\mathbf{C}_y^\Pi \in \mathbb{R}^{N \times M}$ denotes the transportation cost between the $N$ augmented image views and the $M$ textual descriptions of class $j$, which is usually quantified using the cosine similarity. However, adversarial noise breaks alignment with text features, compromising the reliability of similarity measures. As a result, we design the OT cost matrix based our projected features. For image feature $\hat{\mathbf{x}}^n$, we compute:

$$\mathbf{C}_y^\Pi(n,m) = 1 - \cos\left(\Pi(\hat{\mathbf{x}}^n), \mathbf{z}_y^m\right), \tag{8}$$

where $\cos(\cdot, \cdot)$ denotes the cosine similarity. We classify by identifying the class $y$ that yields the lowest transport cost:

$$y = \underset{y \in [K]}{\arg\min}\, d_{\mathrm{OT}}(\mathbb{P}(\mathbf{x}), \mathbb{Q}_y(\mathbf{z}); \mathbf{C}_y^\Pi). \tag{9}$$

Section 3.3 demonstrates that our OT-based classifier achieves larger decision margins, indicating stronger generalization ability.

Table 1: Classification accuracy (%) on 9 widely-used datasets. The best and second best results are highlighted in **bold** and underline, respectively.

| Method | | Pets Clean | Pets Robust | Flowers Clean | Flowers Robust | Aircraft Clean | Aircraft Robust | DTD Clean | DTD Robust | Eurosat Clean | Eurosat Robust | Cars Clean | Cars Robust | Food Clean | Food Robust | SUN Clean | SUN Robust | Caltech101 Clean | Caltech101 Robust |
|---|---|---|---|---|---|---|---|---|---|---|---|---|---|---|---|---|---|---|---|
| PGD Attacks | CLIP | 87.4 | 1.0 | 65.5 | 1.1 | 20.1 | 0.0 | 40.6 | 3.0 | 42.6 | 0.0 | 52.0 | 0.0 | 83.9 | 0.7 | 58.5 | 1.1 | 85.7 | 14.7 |
| | TeCoA | 62.1 | 38.4 | 36.8 | 21.9 | 5.3 | 2.5 | 25.2 | 17.6 | 16.6 | 12.0 | 20.9 | 8.8 | 30.0 | 13.9 | 36.7 | 19.4 | 71.7 | 55.5 |
| | PMG | 65.9 | 41.2 | 37.0 | 23.4 | 5.6 | 2.2 | 21.8 | 15.0 | 18.5 | 12.6 | 25.4 | 11.7 | 36.6 | 18.6 | 38.0 | 22.6 | 75.5 | 61.1 |
| | FARE | 79.4 | 31.1 | 48.0 | 17.1 | 10.9 | 1.4 | 32.1 | 15.6 | 21.9 | 10.7 | 38.7 | 6.8 | 55.3 | 11.7 | 52.4 | 14.9 | 81.0 | 50.7 |
| | RN | 87.4 | 1.9 | 64.6 | 1.5 | 19.2 | 0.0 | 38.0 | 3.7 | 53.2 | 0.2 | 52.1 | 0.2 | 83.4 | 1.2 | 59.7 | 1.7 | 86.6 | 18.9 |
| | TTE | **88.1** | 50.3 | 65.2 | 35.9 | 20.2 | 6.2 | **41.3** | 23.9 | 44.4 | 6.9 | 52.7 | 22.4 | 84.0 | 43.9 | 59.1 | 30.8 | 85.8 | 67.6 |
| | HD | 80.9 | 12.0 | 58.2 | 7.3 | 16.4 | 1.3 | 34.9 | 11.6 | 39.1 | 4.6 | 44.3 | 2.7 | 80.3 | 8.0 | 53.2 | 6.4 | 82.3 | 31.5 |
| | TTC | 83.4 | 57.9 | 64.2 | 39.1 | 18.0 | 13.8 | 37.0 | 27.3 | 53.2 | 12.2 | 48.2 | 33.0 | 82.2 | 57.8 | 55.1 | 41.5 | 86.5 | 65.8 |
| | **COLA** | 87.9 | **77.2** | **66.1** | **50.4** | **20.9** | **15.6** | 41.0 | **34.0** | **53.8** | **19.2** | **54.2** | **35.4** | **84.5** | **63.8** | **61.9** | **45.3** | **88.1** | **75.3** |
| CW Attacks | CLIP | 87.4 | 1.6 | 65.5 | 1.4 | 20.1 | 0.0 | 40.6 | 2.9 | 42.6 | 0.0 | 52.0 | 2.4 | 83.9 | 1.1 | 58.5 | 1.8 | 85.7 | 20.9 |
| | TeCoA | 62.1 | 37.9 | 36.8 | 21.1 | 5.3 | 2.3 | 25.2 | 16.3 | 16.6 | 11.7 | 20.9 | 8.7 | 30.0 | 12.9 | 36.7 | 18.4 | 71.7 | 56.2 |
| | PMG | 65.9 | 39.3 | 37.0 | 21.3 | 5.6 | 1.9 | 21.8 | 13.7 | 18.5 | 11.9 | 25.4 | 10.5 | 36.6 | 16.6 | 38.0 | 20.4 | 75.5 | 61.6 |
| | FARE | 79.4 | 33.9 | 48.0 | 17.3 | 10.9 | 1.4 | 32.1 | 14.4 | 21.9 | 10.7 | 38.7 | 9.1 | 55.3 | 12.9 | 52.4 | 15.7 | 81.0 | 54.9 |
| | RN | 87.4 | 3.1 | 64.6 | 2.1 | 19.2 | 0.0 | 38.0 | 3.5 | 53.2 | 0.2 | 52.1 | 2.4 | 83.4 | 1.9 | 59.7 | 2.5 | 86.6 | 25.9 |
| | TTE | **88.1** | 51.1 | 65.2 | 35.0 | 20.2 | 5.2 | **41.3** | 22.6 | 44.4 | 6.4 | 52.7 | 21.2 | 84.0 | 44.6 | 59.1 | 29.4 | 85.8 | 69.4 |
| | HD | 80.6 | 13.8 | 57.8 | 8.5 | 16.2 | 1.0 | 34.9 | 10.1 | 40.1 | 3.5 | 43.6 | 5.1 | 81.0 | 9.8 | 54.1 | 7.9 | 83.0 | 36.3 |
| | TTC | 83.4 | 57.1 | 64.2 | 36.8 | 18.0 | 12.4 | 37.0 | 27.4 | 53.2 | 12.7 | 48.2 | 30.4 | 82.2 | 54.6 | 55.1 | 39.4 | 86.5 | 66.2 |
| | **COLA** | 87.9 | **63.2** | **66.1** | **41.8** | **20.9** | **15.3** | 41.0 | **31.7** | **53.8** | **13.3** | **54.2** | **35.2** | **84.5** | **54.9** | **61.9** | **40.9** | **88.1** | **72.9** |

## 3.3 Theoretical Analysis

**Global feature alignment preserves pairwise similarity.** We show that projection onto the subspace $\mathcal{U}$ preserves pairwise similarity among adversarial features. Let $\hat{\mathbf{x}}_1$ and $\hat{\mathbf{x}}_2$ be the adversarial counterparts of two clean features $\mathbf{x}_1$ and $\mathbf{x}_2$:

$$\hat{\mathbf{x}}_1 = \mathbf{x}_1 + \boldsymbol{\delta}_1, \quad \hat{\mathbf{x}}_2 = \mathbf{x}_2 + \boldsymbol{\delta}_2, \tag{10}$$

where each perturbation $\boldsymbol{\delta}$ is decomposed as $\boldsymbol{\delta} = \boldsymbol{\delta}_\| + \boldsymbol{\delta}_\perp$, with $\boldsymbol{\delta}_\| \in \mathcal{U}$ and $\boldsymbol{\delta}_\perp \perp \mathcal{U}$. After projection, the adversarial feature satisfies:

$$\Pi(\hat{\mathbf{x}}) = \mathbf{x} + \boldsymbol{\delta}_\|. \tag{11}$$

Let $\Delta$ and $\Delta_\Pi$ denote the deviation from clean similarity before and after projection, respectively:

$$\Delta = |\cos(\hat{\mathbf{x}}_1, \hat{\mathbf{x}}_2) - \cos(\mathbf{x}_1, \mathbf{x}_2)|, \quad \Delta_\Pi = |\cos(\Pi(\hat{\mathbf{x}}_1), \Pi(\hat{\mathbf{x}}_2)) - \cos(\mathbf{x}_1, \mathbf{x}_2)|. \tag{12}$$

We show that projection yields lower cosine similarity distortion, *i.e.*, $\Delta_\Pi \leq \Delta$; the complete proof is given in Appendix A.1.

**Our OT-based framework enjoys larger decision margins.** Recall that the margin of an OT-based classifier with our projected cost matrix for a discrete distribution $\mathbb{P}(\mathbf{x})$ and its label $y$ is defined as:

$$\gamma(\mathbf{C}^\Pi) = \min_{y' \in [K]} d_{\mathrm{OT}}(\mathbb{P}(\mathbf{x}), \mathbb{Q}(\mathbf{z}_{y'}^\Pi); \mathbf{C}_{y'}) - d_{\mathrm{OT}}(\mathbb{P}(\mathbf{x}), \mathbb{Q}(\mathbf{z}_y); \mathbf{C}_y^\Pi), \tag{13}$$

which measures the gap between the OT distance to the true class and the closest competing class. Let $\mathbf{C}_y(n, m) = 1 - \cos(\hat{\mathbf{x}}^n, \mathbf{z}_y^m)$ denote the cost matrix using the original perturbed features, and let $\gamma(\mathbf{C})$ be the corresponding OT classifier margin. We show that the margin of our OT classifier is larger than the original one: $\gamma(\mathbf{C}^\Pi) > \gamma(\mathbf{C})$. The detailed proofs are provided in Appendix A.2. Since classifiers with larger margins imply better generalization [5, 55], our approach leads to improved robustness against adversarial perturbations.

## 4 Experiments

In this section, we present the experimental results of our method under adversarial perturbations, including performance comparisons, ablation studies, and visualization analyses.

Table 2: Classification accuracy (%) on ImageNet and its variants datasets. The best results are highlighted in **bold**.

| Method | | ImageNet | | ImageNet-A | | ImageNet-V2 | | ImageNet-R | | ImageNet-Sketch | | AVG |
|---|---|---|---|---|---|---|---|---|---|---|---|---|
| | | Clean | Robust | Clean | Robust | Clean | Robust | Clean | Robust | Clean | Robust | |
| PGD | CLIP | 62.1 | 1.1 | 30.3 | 0.0 | 55.0 | 0.8 | 65.3 | 6.1 | 39.7 | 5.01 | 26.5 |
| | TTC | 51.7 | 40.0 | 29.6 | 15.4 | 49.3 | 34.4 | 61.4 | 48.5 | 35.1 | 24.4 | 38.9 |
| | **COLA** | **62.8** | **50.0** | **31.8** | **22.7** | **55.4** | **43.2** | **65.7** | **55.6** | **39.4** | **29.8** | **45.6** |
| CW | CLIP | 62.1 | 1.1 | 30.3 | 0.1 | 55.0 | 1.1 | 65.3 | 6.8 | 39.7 | 5.4 | 26.6 |
| | TTC | 51.7 | 38.4 | 29.6 | 13.7 | 49.3 | 31.5 | 61.40 | 46.5 | 35.3 | 30.8 | 38.8 |
| | **COLA** | **62.8** | **42.3** | **37.3** | **15.1** | **55.4** | **34.5** | **65.7** | **49.3** | **40.4** | **33.2** | **43.6** |

Table 3: Classification accuracy (%) on 9 datasets under PGD attacks. The best results are highlighted in **bold**.

| Method | Pets | | Flowers | | Aircraft | | DTD | | Eurosat | | Cars | | Food | | SUN | | Caltech101 | |
|---|---|---|---|---|---|---|---|---|---|---|---|---|---|---|---|---|---|---|
| | Clean | Robust | Clean | Robust | Clean | Robust | Clean | Robust | Clean | Robust | Clean | Robust | Clean | Robust | Clean | Robust | Clean | Robust |
| TeCoA | 62.1 | 38.4 | 36.8 | 21.9 | 5.3 | 2.5 | 25.2 | 17.6 | 16.6 | 12.0 | 20.9 | 8.8 | 30.0 | 13.9 | 36.7 | 19.4 | 71.7 | 55.5 |
| + TTC | 68.0 | 44.1 | 36.7 | 25.1 | 5.5 | 2.9 | 25.2 | 17.9 | 16.6 | 12.7 | 20.4 | 12.0 | 29.9 | 17.8 | 35.4 | 23.9 | 71.7 | 59.2 |
| **+ COLA** | **69.2** | **54.9** | **37.0** | **31.3** | **6.4** | **4.7** | **26.5** | **18.4** | **16.9** | **14.0** | **32.2** | **28.3** | **30.8** | **22.0** | **38.9** | **27.3** | **73.5** | **61.1** |
| PMG | 65.9 | 41.2 | 37.0 | 23.4 | 5.6 | 2.2 | 21.8 | 15.0 | 18.5 | 12.6 | 25.4 | 11.7 | 36.6 | 18.6 | 38.0 | 22.6 | 75.5 | 61.1 |
| + TTC | 63.8 | 43.6 | 36.9 | 26.2 | 5.3 | 2.8 | 21.9 | 16.5 | 18.5 | 14.0 | 25.2 | 14.8 | 36.4 | 21.7 | 36.7 | 25.6 | 75.5 | 63.6 |
| **+ COLA** | **66.0** | **46.2** | **37.8** | **29.0** | **5.8** | **4.5** | **24.5** | **19.0** | **19.7** | **14.9** | **26.9** | **16.8** | **37.9** | **25.1** | **40.8** | **29.1** | **77.5** | **66.6** |
| FARE | 79.4 | 31.1 | 48.0 | 17.1 | 10.9 | 1.4 | 32.1 | 15.6 | 21.9 | 10.7 | 38.7 | 6.8 | 55.3 | 11.7 | 52.4 | 14.9 | 81.0 | 50.7 |
| + TTC | 75.7 | 51.4 | 47.9 | 29.6 | 10.3 | 5.4 | 31.3 | 23.4 | 21.9 | 15.6 | 36.7 | 20.0 | 54.7 | 31.8 | 49.2 | 33.3 | 80.9 | 68.0 |
| **+ COLA** | **80.3** | **57.6** | **48.6** | **35.9** | **11.4** | **7.7** | **33.4** | **26.9** | **22.8** | **16.1** | **41.1** | **28.7** | **55.6** | **35.2** | **54.2** | **45.3** | **83.3** | **73.2** |

## 4.1 Setup

**Datasets.** We evaluate our method on 14 classification datasets spanning a broad range of domains, including generic objects (ImageNet [14], Caltech101 [20]), scenes (SUN397 [58]), textures (DTD [10]), satellite imagery (EuroSAT [23]), and various fine-grained categories such as pets, cars, flowers, food, and aircraft (Pets [39], Cars [26], Flowers [38], Food101 [6], Aircraft [34]). To further assess robustness under distribution shifts, we include five ImageNet variants: ImageNetV2 [46], ImageNet-Sketch [53], ImageNet-A [25], and ImageNet-R [24].

**Implementation details.** The attack budgets, including PDG attack and CW acctack [36, 7], are set of $\epsilon_a = 1/255$ in default. The number of steps for attacks is set as 10. All attacks are bounded by a $L_\infty$ radius. For each test image, we generate $N = 5$ augmented views including the original. For each class, we use the LLM to generate $M = 50$ text descriptions. We select the top-$C = 256$ components from the SVD of class text features to build the projection matrix. All experiments are conducted on a single NVIDIA 3090 GPU if not specified.

**Comparison methods.** Our experiments are based on the pre-trained CLIP model, using ViT-B/32 as the visual encoder and a Transformer as the text encoder. We compare our method with test-time defences including Anti-Adversary (Anti-Adv) [2], Hedge Defence (HD) [57], Test-Time Transformation Ensembling (TTE) [40], and Test-Time Counterattacks (TTC) [59]. These methods are adapted to CLIP without additional networks. We also include fine-tuning-based baselines: TeCoA [36], PMG [54], and FARE [49], which adversarially fine-tune the vision encoder on TinyImageNet [28].

## 4.2 Main Results

**Results on 14 datasets.** We evaluate all methods assuming full access to model weights and gradients by the attacker. Table 1 reports classification accuracy on clean and adversarially perturbed images across 9 diverse datasets. Fine-tuning-based methods such as TeCoA [36], PMG [54], and FARE [49]

Table 4: Classification accuracy (%) on ImageNet and its variants datasets under PGD attacks. The best results are highlighted in **bold**.

| Method | ImageNet Clean | ImageNet Robust | ImageNet-A Clean | ImageNet-A Robust | ImageNet-V2 Clean | ImageNet-V2 Robust | ImageNet-R Clean | ImageNet-R Robust | ImageNet-Sketch Clean | ImageNet-Sketch Robust | AVG |
|---|---|---|---|---|---|---|---|---|---|---|---|
| TeCoA | 36.0 | 19.0 | 6.2 | 1.6 | 30.3 | 15.3 | 38.8 | 23.6 | 16.7 | 10.3 | 19.7 |
| + TTC | 33.9 | 23.8 | 6.1 | 2.8 | 29.1 | 19.5 | 37.6 | 29.9 | 15.9 | 11.3 | 20.9 |
| **+ COLA** | **36.6** | **27.4** | **6.7** | **3.7** | **31.3** | **23.0** | **39.0** | **32.6** | **16.9** | **14.6** | **23.1** |
| PMG | 37.3 | 22.1 | 5.7 | 2.1 | 32.1 | 18.2 | 41.0 | 27.9 | 19.5 | 13.2 | 21.9 |
| + TTC | 35.4 | 25.1 | 5.6 | 3.2 | 31.2 | 20.9 | 40.8 | 33.2 | 18.7 | 18.9 | 23.3 |
| **+ COLA** | **37.8** | **30.2** | **6.3** | **4.3** | **33.1** | **25.2** | **42.4** | **38.3** | 19.3 | **19.9** | **25.6** |
| FARE | 50.4 | 14.0 | 11.7 | 1.0 | 43.0 | 11.3 | 54.8 | 23.1 | 29.3 | 13.5 | 25.2 |
| + TTC | 44.8 | 31.5 | 11.2 | 6.0 | 40.0 | 26.3 | 52.6 | 41.6 | 27.8 | 21.7 | 30.3 |
| **+ COLA** | **50.9** | **37.5** | **11.9** | **7.5** | **44.5** | **29.6** | **55.3** | **45.2** | **29.6** | **25.5** | **33.7** |

Table 5: Accuracy (%) on 9-datasets, ImageNet, and its five variant datasets, evaluated using ViT-B/16 and ViT-L/14 under PGD attacks. The best results are denoted in **bold**.

| Model | ViT-B/16 9-datasets Clean | Robust | ImageNet Clean | Robust | IN-Variants Clean | Robust | ViT-L/14 9-datasets Clean | Robust | ImageNet Clean | Robust | IN-Variants Clean | Robust |
|---|---|---|---|---|---|---|---|---|---|---|---|---|
| CLIP | 63.6 | 0.8 | 67.6 | 0.2 | 57.3 | 1.5 | 70.4 | 3.5 | 73.9 | 2.5 | 69.6 | 4.8 |
| TTC | 61.3 | 13.9 | 67.1 | 20.1 | 53.8 | 17.9 | 69.9 | 16.2 | 68.8 | 21.9 | 68.3 | 19.3 |
| **COLA** | **64.0** | **20.7** | **68.9** | **32.1** | **57.9** | **24.5** | **72.2** | **24.6** | **74.1** | **57.7** | **70.3** | **25.4** |

improve robustness but significantly degrade clean performance. TTC [59], which introduces test-time counterattacks, enhances robustness further but requires stronger counterattack budgets and adds inference complexity.

In contrast, our method consistently improves robustness across all datasets and attack types (PGD and CW), while maintaining competitive clean accuracy. For example, on datasets like Food and Caltech101, our approach achieves over +5% absolute gains in robust accuracy compared to TTC, with only marginal clean performance drops. Table 2 shows results on ImageNet and its challenging variants. Our method consistently outperforms both CLIP and TTC, with especially large robustness gains on ImageNet-A and ImageNet-R—exceeding +7% under PGD attacks.

**Results on finetuned CLIP.** Our method aligns adversarially perturbed image features with their corresponding textual features and can be flexibly integrated into adversarially fine-tuned models as a plug-and-play module, without requiring architectural changes or additional training. Table 3 and Table 4 present results on ImageNet and its variants under PGD attacks.

TTC [59] improves robustness by generating test-time counterattacks using the fine-tuned model, but it often requires stronger counterattack budgets and introduces additional inference overhead. In contrast, our method consistently improves robust accuracy across all settings while preserving or minimally affecting clean performance. Specifically, on the 9-dataset benchmark, our method improves robust accuracy by +16.5% on TeCoA and +5.0% on PMG over their respective baselines, and by +10.8% and +2.6% over their TTC-augmented variants. On more challenging ImageNet variants, our approach achieves the highest robust accuracy in all cases. particularly excelling on ImageNet-R and ImageNet-Sketch, where robustness improvements of over +10% are observed.

**Results on different backbones.** To assess the generality of our method, we evaluate its performance across two CLIP backbones: ViT-B/16 and ViT-L/14. As shown in Table 5, our approach consistently achieves superior robustness against PGD attacks compared to both CLIP and TTC across all datasets. On ViT-B/16, our method improves robust accuracy over TTC by up to 12.0%, with consistent gains across 9-datasets, ImageNet, and its variants. Notably, on ViT-L/14, we observe a substantial 35.8% gain in robust accuracy on ImageNet, alongside improvements on the other test domains. These results highlight the adaptability of our method to different model capacities and architectures, and confirm its effectiveness in enhancing adversarial robustness without compromising clean accuracy.

Table 6: Classification accuracy (%) on 9-datasets under PGD attacks. The best and second best results are highlighted in **bold** and underline, respectively.

| Method | Pets Clean | Pets Robust | Flowers Clean | Flowers Robust | Aircraft Clean | Aircraft Robust | DTD Clean | DTD Robust | Eurosat Clean | Eurosat Robust | Cars Clean | Cars Robust | Food Clean | Food Robust | SUN Clean | SUN Robust | Caltech101 Clean | Caltech101 Robust |
|---|---|---|---|---|---|---|---|---|---|---|---|---|---|---|---|---|---|---|
| CLIP | 87.4 | 0.0 | 65.5 | 0.0 | 20.1 | 0.0 | 40.6 | 0.1 | 42.6 | 0.0 | 52.0 | 0.0 | 83.9 | 0.0 | 58.5 | 0.0 | 85.7 | 0.6 |
| TeCoA | 53.9 | 3.7 | 27.8 | 3.8 | 3.5 | 0.1 | 20.1 | 5.2 | 17.5 | 10.7 | 15.2 | 0.4 | 21.9 | 1.4 | 28.2 | 2.3 | 64.4 | 21.0 |
| PMG | 56.7 | 5.1 | 28.9 | 4.3 | 3.2 | 0.1 | 17.3 | 5.2 | 19.2 | 10.4 | 16.8 | 0.4 | 28.0 | 2.1 | 29.9 | 3.2 | 69.1 | 25.0 |
| FARE | 70.1 | 0.3 | 41.0 | 0.6 | 7.8 | 0.0 | 28.0 | 2.5 | 18.2 | 7.3 | 32.1 | 0.0 | 42.0 | 0.2 | 43.6 | 0.6 | 76.6 | 10.1 |
| RN | 87.4 | 0.0 | 64.6 | 0.0 | 19.2 | 0.0 | 38.0 | 0.1 | 53.2 | 0.0 | 52.1 | 0.0 | 83.4 | 0.0 | 59.7 | 0.0 | 86.6 | 0.7 |
| TTE | **88.1** | 3.2 | 65.2 | 3.5 | 20.2 | 0.4 | **41.4** | 7.2 | 44.4 | 0.1 | 52.7 | 1.5 | 84.0 | 5.3 | 59.1 | 6.0 | 85.8 | 30.2 |
| HD | 80.9 | 0.0 | 58.2 | 0.0 | 16.4 | 0.0 | 34.9 | 0.2 | 39.1 | 0.2 | 44.3 | 0.0 | 80.3 | 0.0 | 53.2 | 0.0 | 82.3 | 1.3 |
| TTC | 64.7 | 24.6 | 63.2 | 13.6 | 16.0 | 6.4 | 35.7 | 11.4 | 53.2 | 13.6 | 41.5 | 12.8 | 80.0 | 17.9 | 46.7 | 13.4 | 86.2 | 36.7 |
| **COLA** | 87.9 | **29.2** | **66.1** | **29.3** | **20.9** | **6.9** | 41.0 | **22.0** | **53.8** | **23.3** | **54.2** | **18.4** | **84.3** | **24.3** | **61.9** | **24.2** | **88.1** | **43.8** |

(Left margin label: $\epsilon_a = 4/255$)

Table 7: Classification accuracy (%) of different models on 9-datasets, ImageNet, and its variant datasets under PGD and CW attacks. The best results are highlighted in **bold**.

| Method | PGD Attacks 9-datasets Clean | Robust | ImageNet Clean | Robust | IN-Variants Clean | Robust | CW Attacks 9-datasets Clean | Robust | ImageNet Clean | Robust | IN-Variants Clean | Robust |
|---|---|---|---|---|---|---|---|---|---|---|---|---|
| CLIP | 59.5 | 2.4 | 62.1 | 1.1 | 47.6 | 3.0 | 59.5 | 3.5 | 62.1 | 1.1 | 47.6 | 3.3 |
| OT w. **C** | 60.7 | 44.9 | 62.5 | 46.3 | 47.9 | 33.2 | 61.6 | 35.2 | 62.3 | 34.5 | 48.0 | 30.7 |
| OT w. $\mathbf{C}^{\Pi}$ | **62.0** | **46.2** | **62.8** | **50.0** | **48.0** | **37.8** | **62.8** | **42.3** | **62.8** | **42.3** | **49.7** | **33.0** |

**Results on large attack budgets.** We further evaluate the robustness of all methods under a stronger adversarial budget of $\epsilon_a = 4/255$. As shown in Table 6, the performance of all baseline models drops sharply, with most robust accuracies approaching zero. This highlights their vulnerability under high-strength attacks. In contrast, our method maintains significantly higher robust accuracy across all nine datasets, demonstrating strong resistance to adversarial degradation. Notably, our approach achieves over 50% absolute gains in robust accuracy on datasets like Food, SUN, and Caltech101 compared to TTC [59], and outperforms all baselines by a large margin under this challenging setting.

## 4.3 Ablation Study

**Effectiveness of the projected cost.** Table 7 presents an ablation study comparing the standard CLIP model, the OT alignment with the original cost matrix $\mathbf{C}$, and our proposed projection-based cost matrix $\mathbf{C}^{\Pi}$. We observe consistent improvements when applying the subspace projection, indicating its effectiveness in mitigating adversarial perturbations. Specifically, across both PGD and CW attacks, $\mathbf{C}^{\Pi}$ achieves higher robust accuracy than both CLIP and the unprojected OT baseline. On the 9-datasets benchmark, the robust accuracy improves from 2.4% (CLIP) to 46.2%, while clean accuracy is also preserved. Similar trends are observed on ImageNet and its variants, with $\mathbf{C}^{\Pi}$ yielding up to over 3% gain in robust accuracy over $\mathbf{C}$ under PGD attakcs.

**Effects of the number of augmentations.** To investigate the sensitivity of our method to augmentation strategies, we evaluate the effect of varying the number of image and class name augmentations on both clean and robust accuracy, as shown in Figure 2. Increasing the number of image augmentations consistently enhances robustness, while the clean accuracy remains stable. However, the improvement becomes marginal when the number exceeds 5. A similar saturation effect is observed in class name augmentation, where performance gains plateau beyond 50 augmentations. These observations indicate that our method is robust to the choice of augmentation hyperparameters.

**Effects of projection matrix construction.** We study how varying the number of singular vectors $C$ used to construct the projection matrix affects classification accuracy. As shown in Figure 3, increasing $C$ steadily improves performance on both clean and adversarial examples across Caltech101 and ImageNet. The gains are more prominent when $C$ is small and gradually saturate beyond $C = 200$,

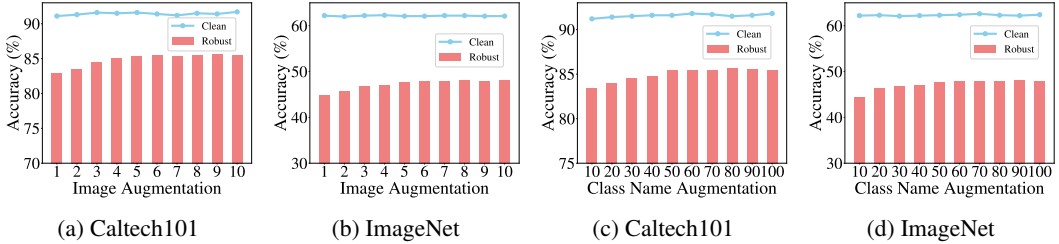

(a) Caltech101 (b) ImageNet (c) Caltech101 (d) ImageNet

Figure 2: Accuracy (%) comparisons across Caltech101 and ImageNet datasets with varying the number of augmentations. (a) and (b): Classification results under different numbers of image augmentations. (c) and (d): Classification results under different numbers of class name augmentations.

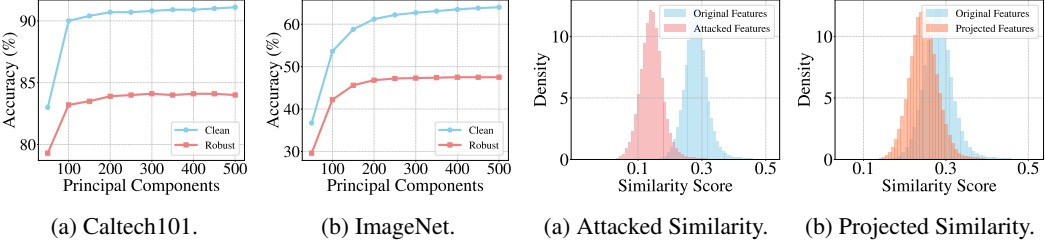

(a) Caltech101. (b) ImageNet. (a) Attacked Similarity. (b) Projected Similarity.

Figure 3: Accuracy (%) with different numbers of principal components in the projection matrix.

Figure 4: Similarity distributions among original, attacked, and projected features on ImageNet.

especially for clean samples. In contrast, robust accuracy improves more slowly, suggesting that while additional components better preserve semantic structure for clean inputs, they provide limited benefit under adversarial perturbations. Based on this observation, we fix $C = 256$ in subsequent experiments to balance performance and efficiency.

To further understand the effectiveness of projection in Eq. (4), we analyze the similarity score distributions between image features and their corresponding text features under different conditions. As shown in Figure 4a, adversarial perturbations significantly reduce the similarity between image and text features, indicating disrupted alignment. In contrast, Figure 4b shows that projecting the attacked features onto the text-induced subspace effectively restores their similarity to the original level. This demonstrates that our projection effectively corrects adversarial misalignment and strengthens semantic consistency across modalities, thereby enhancing robustness in classification.

**Running time.** We compare the inference-time efficiency of our method on ImageNet using CLIP ViT-B/32 with a batch size of 128 on a single NVIDIA 3090 GPU. As shown in Table 8, our method completes evaluation in 28 minutes, significantly faster than TTC (40 minutes) while achieving both higher clean (62.8% vs. 51.7%) and robust (50.0% vs. 40.0%) accuracy. This efficiency stems from the training-free nature of our approach, which avoids the costly iterative optimization required by TTC.

Table 8: Comparison of running time on ImageNet with ViT-B/32.

| Model | Running Time | Accuracy | |
|---|---|---|---|
| | | Clean | Robust |
| CLIP | 10min | 62.1 | 1.1 |
| TTC | 40min | 51.7 | 40.0 |
| **COLA** | 28min | 62.8 | 50.0 |

## 5 Limitation and Conclusion

**Limitation.** While COLA substantially improves adversarial robustness, it still inherits potential biases from the pre-trained vision-language backbone [69, 64, 66]. In particular, the text-induced subspace may encode dataset-specific priors [19, 18], limiting generalization to unseen linguistic or visual domains. Moreover, stronger defenses could provoke more adaptive attacks, suggesting the need for future research on resilience under adaptive adversaries and fairness-aware robustness.

**Conclusion.** By enhancing robustness against adversarial manipulation, COLA contributes to safer multimodal systems, especially in high-stakes applications such as autonomous driving and medical imaging. We present COLA, a training-free and theoretically grounded framework that improves the adversarial robustness of CLIP by addressing modality misalignment. COLA leverages subspace projection to restore global alignment and employs optimal transport to refine local semantic consistency. By embedding projection into the OT cost computation, it maintains cross-modal

alignment without retraining or architectural modification. Theoretical analyses show that COLA reduces cosine distortion and enlarges decision margins, thereby improving generalization. Extensive experiments across 14 benchmarks confirm that COLA consistently enhances zero-shot classification robustness while preserving clean accuracy.

## Acknowledgments and Disclosure of Funding

This research is supported by the National Natural Science Foundation of Anhui (Grant No. 2508085MF143) and the advanced computing resources provided by the Supercomputing Center of the University of Science and Technology of China (USTC). Additional support was provided by the National Research Foundation, Singapore, under the NRF Investigatorship Award (NRF-NRFI10-2024-0004).

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

# Contents

# A Proofs

## A.1 Proof of Cosine Similarity Distortion Bound

Let $\mathbf{x}_1, \mathbf{x}_2 \in \mathbb{R}^d$ be clean feature vectors with $\|\mathbf{x}_i\| = 1$. The adversarial features are denoted by

$$\hat{\mathbf{x}}_i = \mathbf{x}_i + \boldsymbol{\delta}_i, \quad \|\boldsymbol{\delta}_i\| \leq \epsilon,$$

where $\boldsymbol{\delta}_i = \boldsymbol{\delta}_\parallel^i + \boldsymbol{\delta}_\perp^i$, with $\boldsymbol{\delta}_\parallel^i \in \mathcal{U}$ and $\boldsymbol{\delta}_\perp^i \perp \mathcal{U}$. Let $\Pi(\hat{\mathbf{x}}_i) = \mathbf{x}_i + \boldsymbol{\delta}_\parallel^i$ denote the projection of the adversarial feature onto the subspace $\mathcal{U}$. The cosine similarity between adversarial features and projected features:

$$\cos(\hat{\mathbf{x}}_1, \hat{\mathbf{x}}_2) = \frac{(\mathbf{x}_1 + \boldsymbol{\delta}_1)^\top (\mathbf{x}_2 + \boldsymbol{\delta}_2)}{\|\hat{\mathbf{x}}_1\| \cdot \|\hat{\mathbf{x}}_2\|}, \quad \cos(\Pi(\hat{\mathbf{x}}_1), \Pi(\hat{\mathbf{x}}_2)) = \frac{(\mathbf{x}_1 + \boldsymbol{\delta}_\parallel^1)^\top (\mathbf{x}_2 + \boldsymbol{\delta}_\parallel^2)}{\|\Pi(\hat{\mathbf{x}}_1)\| \cdot \|\Pi(\hat{\mathbf{x}}_2)\|}.$$

We define the cosine distortion quantities as

$$\Delta = |\cos(\hat{\mathbf{x}}_1, \hat{\mathbf{x}}_2) - \cos(\mathbf{x}_1, \mathbf{x}_2)|, \quad \Delta_\Pi = |\cos(\Pi(\hat{\mathbf{x}}_1), \Pi(\hat{\mathbf{x}}_2)) - \cos(\mathbf{x}_1, \mathbf{x}_2)|.$$

We now compare $\Delta$ and $\Delta_\Pi$. Using a second-order Taylor approximation for small perturbations $\|\boldsymbol{\delta}_i\| \ll 1$, we obtain:

$$\cos(\hat{\mathbf{x}}_1, \hat{\mathbf{x}}_2) \approx (\mathbf{x}_1^\top \mathbf{x}_2 + \mathbf{x}_1^\top \boldsymbol{\delta}_2 + \boldsymbol{\delta}_1^\top \mathbf{x}_2 + \boldsymbol{\delta}_1^\top \boldsymbol{\delta}_2)(1 - \mathbf{x}_1^\top \boldsymbol{\delta}_1 - \frac{1}{2}\|\boldsymbol{\delta}_1\|^2)(1 - \mathbf{x}_2^\top \boldsymbol{\delta}_2 - \frac{1}{2}\|\boldsymbol{\delta}_2\|^2).$$

$$\cos(\Pi(\hat{\mathbf{x}}_1), \Pi(\hat{\mathbf{x}}_2)) \approx (\mathbf{x}_1^\top \mathbf{x}_2 + \mathbf{x}_1^\top \boldsymbol{\delta}_\parallel^2 + (\boldsymbol{\delta}_\parallel^1)^\top \mathbf{x}_2 + (\boldsymbol{\delta}_\parallel^1)^\top \boldsymbol{\delta}_\parallel^2)$$
$$\times (1 - \mathbf{x}_1^\top \boldsymbol{\delta}_\parallel^1 - \frac{1}{2}\|\boldsymbol{\delta}_\parallel^1\|^2)(1 - \mathbf{x}_2^\top \boldsymbol{\delta}_\parallel^2 - \frac{1}{2}\|\boldsymbol{\delta}_\parallel^2\|^2).$$

To simplify analysis, assume $\boldsymbol{\delta}_1 = \boldsymbol{\delta}_2 = \boldsymbol{\delta}$ and $\boldsymbol{\delta}_\parallel^1 = \boldsymbol{\delta}_\parallel^2 = \boldsymbol{\delta}_\parallel$. Since $\mathbf{x}_i \in \mathcal{U}$ and $\boldsymbol{\delta}_\perp \perp \mathcal{U}$, we have $\mathbf{x}_i^\top \boldsymbol{\delta}_\perp = 0$ and thus $\mathbf{x}_i^\top \boldsymbol{\delta} = \mathbf{x}_i^\top \boldsymbol{\delta}_\parallel$.

Under this setting, both distortions simplify to:

$$\Delta \approx 2\mathbf{x}_1^\top \boldsymbol{\delta}_\parallel - 2(\mathbf{x}_1^\top \mathbf{x}_2)(\mathbf{x}_1^\top \boldsymbol{\delta}_\parallel) + \mathcal{O}(\epsilon^2),$$

$$\Delta_\Pi \approx 2\mathbf{x}_1^\top \boldsymbol{\delta}_\parallel - 2(\mathbf{x}_1^\top \mathbf{x}_2)(\mathbf{x}_1^\top \boldsymbol{\delta}_\parallel) + \mathcal{O}(\epsilon^2).$$

Thus, the first-order terms in $\Delta$ and $\Delta_\Pi$ are identical. However, in the general case where $\|\boldsymbol{\delta}_\perp\| > 0$, the norm of the full feature $\hat{\mathbf{x}}_i$ is larger than that of its projection $\Pi(\hat{\mathbf{x}}_i)$, reducing the cosine similarity in the unprojected case.

Using Cauchy-Schwarz and bounding terms:

$$|\Delta_\Pi| \leq 2\|\boldsymbol{\delta}_\parallel\|(1 + |\mathbf{x}_1^\top \mathbf{x}_2|), \quad |\Delta| \geq \frac{2\|\boldsymbol{\delta}_\parallel\|(1 + |\mathbf{x}_1^\top \mathbf{x}_2|)}{\sqrt{1 + \frac{\|\boldsymbol{\delta}_\perp\|^2}{\|\boldsymbol{\delta}_\parallel\|^2}}}.$$

Therefore, when the perturbation contains a non-zero orthogonal component ($\|\boldsymbol{\delta}_\perp\| > 0$), we have

$$\frac{\Delta_\Pi}{\Delta} \leq \sqrt{\frac{\|\boldsymbol{\delta}_\parallel\|^2}{\|\boldsymbol{\delta}\|^2}} < 1,$$

which shows that the cosine similarity distortion is strictly reduced by projecting the adversarial features onto the subspace $\mathcal{U}$.

**General case.** When $\boldsymbol{\delta}_1 \neq \boldsymbol{\delta}_2$, let $s = \mathbf{x}_1^\top \mathbf{x}_2$ and $s_\parallel = \Pi(\hat{\mathbf{x}}_1)^\top \Pi(\hat{\mathbf{x}}_2)$. A first-order Taylor expansion yields:

$$\cos(\hat{\mathbf{x}}_1, \hat{\mathbf{x}}_2) \approx s + \mathbf{x}_1^\top \boldsymbol{\delta}_2 + \mathbf{x}_2^\top \boldsymbol{\delta}_1 - s(\mathbf{x}_1^\top \boldsymbol{\delta}_1 + \mathbf{x}_2^\top \boldsymbol{\delta}_2),$$

$$\cos(\Pi(\hat{\mathbf{x}}_1), \Pi(\hat{\mathbf{x}}_2)) \approx s_\parallel + \mathbf{x}_1^\top \boldsymbol{\delta}_{2\parallel} + \mathbf{x}_2^\top \boldsymbol{\delta}_{1\parallel} - s_\parallel(\mathbf{x}_1^\top \boldsymbol{\delta}_{1\parallel} + \mathbf{x}_2^\top \boldsymbol{\delta}_{2\parallel}).$$

Hence, the projected distortion satisfies:

$$\Delta_\Pi \le |s - s_\||| + \epsilon \left( \frac{1}{\|\Pi \mathbf{x}_1\|} + \frac{1}{\|\Pi \mathbf{x}_2\|} \right),$$

where the first term reflects the semantic deviation between clean and projected subspaces, and the $\epsilon$ term accounts for asymmetric perturbations. Since projection removes orthogonal noise, the overall distortion ratio becomes:

$$\frac{\Delta_\Pi}{\Delta} \lesssim \frac{1}{1 + |s|} < 1,$$

showing that projection consistently suppresses cosine similarity distortion even when perturbations differ in direction or magnitude.

## A.2  Proof of OT-Margin Amplification

We prove that the projected OT margin satisfies $\gamma(\mathbf{C}^\Pi) \ge \gamma(\mathbf{C})$, as stated in the main text.

*Proof.* Since the projection $\Pi(\hat{\mathbf{x}}^n) = \mathbf{x}^n + \boldsymbol{\delta}_\|^n$ removes the orthogonal perturbation $\boldsymbol{\delta}_\perp^n \perp \mathcal{U}$, and each text prototype $\mathbf{z}_y^m$ lies in the subspace $\mathcal{U}$, the dot product is preserved:

$$(\hat{\mathbf{x}}^n)^\top \mathbf{z}_y^m = \left( \Pi(\hat{\mathbf{x}}^n) \right)^\top \mathbf{z}_y^m.$$

Meanwhile, the norm of the adversarial feature satisfies:

$$\|\hat{\mathbf{x}}^n\| = \sqrt{\|\Pi(\hat{\mathbf{x}}^n)\|^2 + \|\boldsymbol{\delta}_\perp^n\|^2} \ge \|\Pi(\hat{\mathbf{x}}^n)\|,$$

with equality if and only if $\boldsymbol{\delta}_\perp^n = 0$. Therefore, the cosine similarities before and after projection are:

$$\cos(\hat{\mathbf{x}}^n, \mathbf{z}_y^m) = \frac{\left( \Pi(\hat{\mathbf{x}}^n) \right)^\top \mathbf{z}_y^m}{\|\hat{\mathbf{x}}^n\| \|\mathbf{z}_y^m\|}, \quad \cos(\Pi(\hat{\mathbf{x}}^n), \mathbf{z}_y^m) = \frac{\left( \Pi(\hat{\mathbf{x}}^n) \right)^\top \mathbf{z}_y^m}{\|\Pi(\hat{\mathbf{x}}^n)\| \|\mathbf{z}_y^m\|}.$$

Since $\|\hat{\mathbf{x}}^n\| \ge \|\Pi(\hat{\mathbf{x}}^n)\|$, we conclude:

$$\cos(\Pi(\hat{\mathbf{x}}^n), \mathbf{z}_y^m) = \cos(\hat{\mathbf{x}}^n, \mathbf{z}_y^m) \cdot \frac{\|\hat{\mathbf{x}}^n\|}{\|\Pi(\hat{\mathbf{x}}^n)\|} \ge \cos(\hat{\mathbf{x}}^n, \mathbf{z}_y^m).$$

Hence, the projected cost matrix entry is smaller:

$$\mathbf{C}_y^\Pi(n, m) = 1 - \cos(\Pi(\hat{\mathbf{x}}^n), \mathbf{z}_y^m) \le 1 - \cos(\hat{\mathbf{x}}^n, \mathbf{z}_y^m) = \mathbf{C}_y(n, m).$$

Given $\mathbf{C}_y^\Pi(n, m) \le \mathbf{C}_y(n, m)$, we now compare the OT distances. The OT distance is defined as:

$$d_{\mathrm{OT}}(\mathbb{P}, \mathbb{Q}; \mathbf{C}_y) = \min_{\mathbf{T}_y \ge 0} \langle \mathbf{T}_y, \mathbf{C}_y \rangle, \quad \text{s.t. } \mathbf{T}_y \mathbf{1}_M = \mathbf{a}, \quad \mathbf{T}_y^\top \mathbf{1}_N = \mathbf{b}_y.$$

For any feasible transport plan $\mathbf{T}_y$, we have $\langle \mathbf{T}_y, \mathbf{C}_y^\Pi \rangle \le \langle \mathbf{T}_y, \mathbf{C}_y \rangle$. Let $\mathbf{T}_y^* = \arg\min \langle \mathbf{T}_y, \mathbf{C}_y \rangle$. Then:

$$d_{\mathrm{OT}}(\mathbb{P}, \mathbb{Q}; \mathbf{C}_y^\Pi) \le \langle \mathbf{T}_y^*, \mathbf{C}_y^\Pi \rangle \le \langle \mathbf{T}_y^*, \mathbf{C}_y \rangle = d_{\mathrm{OT}}(\mathbb{P}, \mathbb{Q}; \mathbf{C}_y).$$

Define the OT distance reduction:

$$\Delta d_y = d_{\mathrm{OT}}(\mathbb{P}, \mathbb{Q}; \mathbf{C}_y) - d_{\mathrm{OT}}(\mathbb{P}, \mathbb{Q}; \mathbf{C}_y^\Pi) \ge 0.$$

Next, we compare the cost reductions for the true class $y^\star$ and a competing class $y_{\mathrm{neg}}$. Note that:

$$\mathbf{C}_y(n, m) - \mathbf{C}_y^\Pi(n, m) = \cos(\Pi(\hat{\mathbf{x}}^n), \mathbf{z}_y^m) - \cos(\hat{\mathbf{x}}^n, \mathbf{z}_y^m) = \cos(\hat{\mathbf{x}}^n, \mathbf{z}_y^m) \cdot \left( \frac{\|\hat{\mathbf{x}}^n\|}{\|\Pi(\hat{\mathbf{x}}^n)\|} - 1 \right).$$

The multiplicative factor $\frac{\|\hat{\mathbf{x}}^n\|}{\|\Pi(\hat{\mathbf{x}}^n)\|} - 1 \ge 0$ is shared across all classes. Since $\cos(\hat{\mathbf{x}}^n, \mathbf{z}_{y^\star}^m) \ge \cos(\hat{\mathbf{x}}^n, \mathbf{z}_{y_{\mathrm{neg}}}^m)$, it follows that:

$$\mathbf{C}_{y^\star}(n, m) - \mathbf{C}_{y^\star}^\Pi(n, m) \ge \mathbf{C}_{y_{\mathrm{neg}}}(n, m) - \mathbf{C}_{y_{\mathrm{neg}}}^\Pi(n, m).$$

Table 9: Performance comparison under AutoAttack.

| Model | 9-datasets (Clean) | 9-datasets (Robust) | ImageNet (Clean) | ImageNet (Robust) |
|-------|--------------------|--------------------|--------------------|--------------------|
| CLIP | 63.6 | 0.5 | 67.6 | 0.2 |
| TTC | 61.3 | 8.3 | 67.1 | 16.2 |
| **COLA** | **64.0** | **18.9** | **68.9** | **29.6** |

---

**Algorithm 1** Pipeline of **COLA**

---

**Input:** Adversarial visual feature $\mathbf{x}$ with its $N$ augmentations $\{\hat{\mathbf{x}}^n\}_{n=1}^N$, and class textual embeddings $\{\mathbf{z}_y^m\}_{y,m}$ from LLM-generated descriptions.

**Global feature projection.** Construct the textual embedding matrix $\mathbf{Z} \in \mathbb{R}^{d \times KM}$ and perform SVD: $\mathbf{Z} = \mathbf{U}\boldsymbol{\Sigma}\mathbf{V}^\top$. Extract top-$C$ principal components $\mathbf{U}_C = \mathbf{U}_{[:,1:C]}$ and project each adversarial image feature onto the text-induced subspace: $\Pi(\hat{\mathbf{x}}^n) = \mathbf{U}_C \mathbf{U}_C^\top \hat{\mathbf{x}}^n$.

**Local semantic distribution modeling.** Represent the image and class as discrete distributions: $P(\mathbf{x}) = \sum_n a^n \delta(\hat{\mathbf{x}}^n - \mathbf{x})$, $Q_y(\mathbf{z}) = \sum_m b_y^m \delta(\mathbf{z}_y^m - \mathbf{z})$. The importance weights are normalized by prediction entropy: $a^n \propto \exp(h(\hat{\mathbf{x}}^n))$, $b_y^m \propto \exp(h(\mathbf{z}_y^m))$, where $h(\cdot)$ is defined in Eq. (6).

**Optimal transport alignment.** For each image–text pair $(\hat{\mathbf{x}}^n, \mathbf{z}_y^m)$, compute the projected transport cost: $C_y^\Pi(n,m) = 1 - \cos(\Pi(\hat{\mathbf{x}}^n), \mathbf{z}_y^m)$. Obtain the OT distance by solving:

$$d_{\mathrm{OT}}\big(P(\mathbf{x}), Q_y(\mathbf{z}); C_y^\Pi\big) = \min_{\mathbf{T}_y \geq 0} \langle \mathbf{T}_y, C_y^\Pi \rangle, \text{ s.t. } \mathbf{T}_y \mathbf{1}_M = a, \ \mathbf{T}_y^\top \mathbf{1}_N = b_y.$$

**Classification.** Predict the label with minimal OT distance:

$$\hat{y} = \arg \min_{y \in [K]} d_{\mathrm{OT}}\big(P(\mathbf{x}), Q_y(\mathbf{z}); C_y^\Pi\big).$$

---

Hence, for the respective optimal transport plans $\mathbf{T}_{y^\star}^*$ and $\mathbf{T}_{y_{\mathrm{neg}}}^*$, we have:

$$\langle \mathbf{T}_{y^\star}^*, \mathbf{C}_{y^\star} - \mathbf{C}_{y^\star}^\Pi \rangle \geq \langle \mathbf{T}_{y_{\mathrm{neg}}}^*, \mathbf{C}_{y_{\mathrm{neg}}} - \mathbf{C}_{y_{\mathrm{neg}}}^\Pi \rangle.$$

Therefore:

$$\Delta d_{y^\star} \geq \Delta d_{y_{\mathrm{neg}}}.$$

Finally, we analyze the margin difference:

$$\gamma(\mathbf{C}^\Pi) - \gamma(\mathbf{C}) = \big[d_{\mathrm{OT}}(\mathbb{P}, \mathbb{Q}; \mathbf{C}_{y_{\mathrm{neg}}}^\Pi) - d_{\mathrm{OT}}(\mathbb{P}, \mathbb{Q}; \mathbf{C}_{y_{\mathrm{neg}}})\big] - \big[d_{\mathrm{OT}}(\mathbb{P}, \mathbb{Q}; \mathbf{C}_{y^\star}^\Pi) - d_{\mathrm{OT}}(\mathbb{P}, \mathbb{Q}; \mathbf{C}_{y^\star})\big]$$

$$= -\Delta d_{y_{\mathrm{neg}}} + \Delta d_{y^\star} = \Delta d_{y^\star} - \Delta d_{y_{\mathrm{neg}}} \geq 0.$$

Thus, $\gamma(\mathbf{C}^\Pi) \geq \gamma(\mathbf{C})$, with equality if and only if $\boldsymbol{\delta}_\perp^n = 0$ for all $n$. $\qquad\square$

## B  External Results

**Analysis.** As shown in Table 9, our method achieves the best robustness under *AutoAttack*, reaching 18.9% on 9-datasets and 29.6% on ImageNet, far surpassing CLIP (0.5% / 0.2%) and TTC (8.3% / 16.2%). These results show that the proposed subspace projection effectively filters non-semantic adversarial noise, while OT-based alignment restores image–text consistency.

## C  Algorithm

The overall procedure of COLA is summarized in Algorithm 1, which outlines the projection-based alignment and OT-based matching steps for adversarially robust inference.

