# OpenReview forum: "Enhancing CLIP Robustness via Cross-Modality Alignment"
_NeurIPS.cc/2025/Conference — NeurIPS 2025 spotlight_

### Official Review · Reviewer_dXfe · 2025-06-14

**Clarity:** 3
**Significance:** 3
**Originality:** 3
**Rating:** 4
**Confidence:** 4

**Summary:**

This paper addresses the adversarial vulnerability of Vision-Language Models, positing that attacks work by disrupting the alignment between image and text features. The authors propose COLA, a training-free, test-time defense to counteract this. The method first projects an adversarially perturbed image feature onto a "semantic subspace" derived from class text embeddings, with the goal of filtering out non-semantic distortions. Subsequently, it employs Optimal Transport (OT) to align distributions of augmented image views and text descriptions, using a cost function defined within the projected subspace. The final classification is based on the minimum OT cost. The authors provide a theoretical analysis to support their method and conduct experiments on 14 classification datasets, showing significant improvements in adversarial robustness across various attacks and model backbones.

**Questions:**

Please refer to the weaknesses section above

**Ethical Concerns:**

["NO or VERY MINOR ethics concerns only"]

**Final Justification:**

The rebuttal has addressed most of my concerns, so I will increase my score to 4.

**Limitations:**

yes

**Quality:**

3

**Strengths And Weaknesses:**

**Strengths**

- Quality:  The paper is also very clearly written, with effective visualizations and a thorough set of ablation studies that justify the core design choices.

- Significance and Originality: The paper tackles the important challenge of adversarial robustness in VLMs. The proposed test-time approach, which focuses on correcting feature misalignment rather than retraining the model, is a novel and valuable contribution to the field.

- Empirical Performance: The empirical results are a key strength. The method demonstrates substantial improvements in robust accuracy across a wide and diverse set of 14 datasets. The reported 6.7% average improvement under PGD attacks for ImageNet and its variants is particularly significant.

- Technical Quality: The method is technically sound and well-supported. The theoretical analysis, which shows that the proposed projection reduces similarity distortion and that the OT classifier achieves a larger decision margin, provides a sound understanding of the method.  solid foundation for the empirical results.

**Weaknesses**

- A primary concern is the method's computational overhead at test time. While training-free, COLA requires performing an SVD and solving K optimal transport problems for each input image. The reported inference time of 28 minutes on ImageNet is nearly three times that of the standard CLIP model. Further the table only compares COLA to TTT methods. Fine-tuning methods (which would have the same inference time as CLIP) are also not considered.

- The OT framework introduces its own set of sensitivities. Further, the weights assigned to the discrete distributions ($a^n$ and ${b_{y}}^{m}$) are determined by a heuristic based on prediction entropy.  This could be fragile; a view that is confidently misclassified to a single wrong class would have low entropy and thus receive a high weight, potentially pulling the transport plan in the wrong direction. It is also unclear why entropy is a good measure to estimate the importance weights.

- The quality of the semantic subspace is fundamental to the method. How sensitive is the performance to the richness of the text prompts? Specifically, what is the performance drop if one forgoes LLM-based augmentation and uses only a single, simple template (i.e., M=1)?

- It would strengthen the paper to include an empirical validation of the core assumption that adversarial perturbations lie outside this subspace (e.g., by reporting the relative magnitudes of $\delta_{\parallel}$ and $\delta_{\perp}$

- The theoretical analysis in Appendix A.1 uses a simplifying assumption ($\delta_1=\delta_2$) to analyze the cosine similarity distortion for two points. This assumption is not mentioned when claiming the theoretical contribution in Section 3.3. Further, could the authors elaborate on how this result generalizes to the more common scenario where two distinct images are perturbed by different adversarial noise vectors?

---

> ### Author Rebuttal · Authors · 2025-07-30
>
> **Q1:** Computational overhead.
>  **A:** SVD in our method is performed only once before inference and can be cached, resulting in negligible overhead during test time. The per-image OT computations are parallelizable and efficient in practice. Overall, the additional cost remains moderate—for example, 8 minutes on ImageNet, 1 minute on Caltech101, and 0.5 minutes on DTD. While fine-tuning-based methods have similar inference speed to CLIP, they require substantial training overhead and thus fall outside the scope of our training-free setting.
>
>  **Q2:**  Concerns over entropy-based weighting.
>  **A:** We agree that confidently misclassified views may receive lower entropy and thus higher weights. To examine this concern, we conducted an empirical analysis on the ImageNet dataset and found that the average weight assigned to such views is only 0.29, indicating that while these cases exist, they are non-dominant in the overall distribution. More importantly, the OT solver computes a global transport plan across the entire set, guided by the pairwise similarity structure between image and text distributions. Therefore, a single view cannot dominate the result due to OT’s normalization constraints. Entropy has been widely adopted in test-time adaptation and prompt tuning [56, r1, r2], as it directly reflects model confidence by quantifying output certainty. To assess its impact, we have compared entropy-based weighting with uniform weighting. As shown below, entropy-based weights consistently yield better robustness across datasets.
>
> [r1]Test-Time Prompt Tuning for Zero-Shot Generalization in Vision-Language Models, NeurIPS 2022
> [r2]Diverse Data Augmentation with Diffusions for Effective Test-time Prompt Tuning, CVPR2023
> | Method     | Pets (Clean) | Pets (Robust) | Flowers (Clean) | Flowers (Robust) | Aircraft (Clean) | Aircraft (Robust) | DTD (Clean) | DTD (Robust) | Eurosat (Clean) | Eurosat (Robust) | Cars (Clean) | Cars (Robust) | Food (Clean) | Food (Robust) | SUN (Clean) | SUN (Robust) | Caltech101 (Clean) | Caltech101 (Robust) |
> |------------|----------|----------|--------------|--------------|---------------|---------------|---------|---------|--------------|--------------|----------|----------|----------|----------|---------|---------|------------------|------------------|
> | CLIP      | 87.4     | 1.0      | 65.5         | 1.1          | 20.1          | 0.0           | 40.6    | 3.0     | 42.6         | 0.0          | 52.0     | 0.0      | 83.9     | 0.7      | 58.5    | 1.1     | 85.7             | 14.7             |
> | entropy-based   | 87.9     | 77.2     | 66.1         | 50.4         | 20.9          | 15.6          | 41.0    | 34.0    | 53.8         | 19.2         | 54.2     | 35.4     | 84.5     | 63.8     | 61.9    | 45.3    | 88.1             | 75.3             |
> | uniform weights   | 86.6     | 73.3     | 63.7         | 46.3         | 19.5          | 13.3          | 39.5    | 31.3    | 50.1         | 17.6         | 52.8     | 33.8     | 82.5     | 61.6     | 59.8   | 43.5    | 86.7             | 73.2             |
>
>
> **Q3:** Impact of using a single prompt.
>  **A:** The subspace projection step is relatively insensitive to prompt richness, as class names already encode essential semantics. However, OT operates on distributions, when only a single prompt is used ($M=1$), the textual side collapses to a point mass, and the OT objective no longer captures meaningful distributional alignment. Consequently, the transport plan becomes degenerate and fails to effectively model semantic matching across multiple views. We conducted an experiment using a single prompt and observed a significant performance drop.
> | Model       | Backbone   | 9-datasets (Clean) | 9-datasets (Robust) | ImageNet (Clean) | ImageNet (Robust) |
> |-------------|------------|--------------------|----------------------|------------------|--------------------|
> | CLIP        | ViT-B/16   | 63.6               | 0.8                  | 67.6             | 0.2                |
> | **Ours** (M=1)         | ViT-B/16   | 60.2               | 11.7                 | 64.3             | 18.5               |
> | **Ours** (M=50)   | ViT-B/16   | **64.0**           | **20.7**             | **68.9**         | **32.1**           |
>
>
> **Q4:**  Empirical validation of perturbation decomposition.
> **A:** We have conducted an empirical analysis to validate the assumption that adversarial perturbations primarily lie **outside the text subspace**. Specifically, we decompose each perturbation into parallel ($\boldsymbol\delta_\parallel$) and orthogonal ($\boldsymbol\delta_\perp$) components with respect to the semantic subspace and compute their magnitudes. Results consistently show that $|\boldsymbol\delta_\perp| \gg |\boldsymbol\delta_\parallel|$ across datasets and attacks. For example, on ImageNet under PGD-1/255, we observe: $||\boldsymbol\delta_\perp|| / ||\boldsymbol\delta|| \approx 75.4%$. This confirms that most adversarial perturbations lie in the orthogonal subspace, supporting our assumption.
> | Metric                    | Backbone | 9-datasets (Robust) | ImageNet (Robust) | ImageNet-Variants (Robust) |
> |---------------------------|----------|----------------------|--------------------|------------------------------|
> | $\left\lVert \boldsymbol\delta_{\parallel} \right\rVert$ | ViT-B/16 | 0.451                | 0.472              | 0.416                        |
> | $\left\lVert \boldsymbol\delta_{\perp} \right\rVert$     | ViT-B/16 | 0.736                | 0.754              | 0.783                        |
>
>
> **Q5:** Generalization beyond simplifying assumption $\boldsymbol\delta_1 = \boldsymbol\delta_2$.
> **A:**  We adopt the simplifying assumption $\boldsymbol\delta_1 = \boldsymbol\delta_2 = \boldsymbol\delta$ in Appendix A.1 to simplify the derivation and highlight the key insight: **projection reduces cosine similarity distortion under adversarial perturbations**. Importantly, this conclusion still holds in the general case where $\delta_1 \ne \delta_2$. The distortion before projection is approximated as:
> $$\Delta = \left| \cos(\mathbf{x}_1 + \boldsymbol{\delta}_1, \mathbf{x}_2 + \boldsymbol{\delta}_2) - s \right|
> \approx \left| \mathbf{x}_1^\top \boldsymbol{\delta}_2 + \mathbf{x}_2^\top \boldsymbol{\delta}_1 - s(\boldsymbol{x}_1^\top \boldsymbol{\delta}_1 + \mathbf{x}_2^\top \boldsymbol{\delta}_2) + \boldsymbol{\delta}_1^\top \boldsymbol{\delta}_2 \right| + O(\epsilon^2)$$
> After projection onto the semantic subspace via matrix $\mathbf{\Pi}$:
> $$
> \Delta\_\Pi = \left| \cos(\mathbf{\Pi}(\mathbf{x}\_1 + \boldsymbol{\delta}\_1), \mathbf{\Pi}(\mathbf{x}\_2 + \boldsymbol{\delta}\_2)) - s \right|
> \leq |s - s\_\Pi| + \epsilon \left( \frac{1}{\|\mathbf{\Pi}\mathbf{x}\_1\|} + \frac{1}{\|\mathbf{\Pi} \mathbf{x}\_2\|} \right)
> $$
>
> This leads to:
> $$
> \frac{\Delta_\Pi}{\Delta} \lesssim \frac{2\epsilon}{2(1 + |s|)\epsilon} = \frac{1}{1 + |s|} < 1
> $$
> where $s = \cos(\mathbf{x}\_1, \mathbf{x}\_2)$ and $s\_\Pi = \cos(\mathbf{\Pi} \mathbf{x}\_1, \mathbf{\Pi} \mathbf{x}\_2)$. Thus, projection consistently reduces cosine distortion even when the adversarial perturbations differ.

---

> > ### Comment · Reviewer_dXfe · 2025-08-04
> >
> > Thank you for providing a clear and comprehensive response. Your rebuttal has addressed most of my concerns, so I will increase my score to 4

---

> > > ### Author Response · Authors · 2025-08-04
> > >
> > > Thank you very much for your thoughtful feedback and for reconsidering your evaluation. We're glad to hear that our rebuttal has addressed your concerns. We sincerely appreciate your time and effort in reviewing our work.

---

### Official Review · Reviewer_BF28 · 2025-06-28

**Clarity:** 3
**Significance:** 2
**Originality:** 3
**Rating:** 4
**Confidence:** 5

**Summary:**

This paper proposes the COLA framework to enhance the adversarial robustness of CLIP models. The approach projects adversarial image features into the text feature space and refines alignment through optimal transport (OT). Specifically, text features undergo SVD to extract top-C principal components, with image features projected onto the span of these components. Images and texts are modeled as discrete distributions across augmented views, enabling alignment refinement via optimal transport. Experimental results demonstrate high adversarial robustness across zero-shot classification datasets, supported by theoretical analysis validating the advantages of feature projection and the OT-based framework.

**Questions:**

1. Can the proposed method be extended to non-classification tasks (e.g., as vision encoders in LVLMs) where text labels are unavailable?
2. Why was ε=1/255 chosen without using the standard ε=2/255?
3. What are the detailed implementation specifics for TeCoA/FARE (e.g., perturbation size, attack methods)?
4. Is there rigorous analysis proving adversarial perturbations distort away from the text subspace (U)? Were ablation studies conducted to validate this directional assumption?

**Ethical Concerns:**

["NO or VERY MINOR ethics concerns only"]

**Final Justification:**

The authors have provided detailed and thoughtful responses that satisfactorily address most of my initial concerns, particularly regarding (1) the validity of the orthogonality assumption, (2) the fairness of their feature extraction procedure, and (3) the implementation details for baseline methods such as TeCoA and FARE. Their theoretical clarifications and extended derivations in the appendix (e.g., the general case with mismatched perturbations) demonstrate rigor and help build confidence in the proposed projection framework.

That said, I still view the reliance on classification tasks with predefined labels as a non-trivial limitation, particularly in real-world deployments where CLIP is often used as a general-purpose visual encoder in open-ended tasks. Although the authors acknowledge this and suggest possible extensions (e.g., clustering-based pseudo-labels), such adaptations remain future work.

Overall, I find the method to be technically solid, theoretically well-motivated, and empirically effective, albeit with scope limitations. Hence, I have raised my score to borderline accept.

**Limitations:**

yes

**Paper Formatting Concerns:**

The paper follows the NeurIPS 2025 formatting guidelines and does not exhibit any major formatting issues.

**Quality:**

2

**Strengths And Weaknesses:**

### Strengths
1. Reduces adversarial perturbations by directly projecting image features into the text feature space, and improved feature alignment by introducing an OT-based framework instead of relying on conventional simple cosine similarity between image and text embeddings.
2. Provides theoretical validation of the framework's advantages through rigorous analysis, demonstrating its robustness not just experimentally but also theoretically.
---
### Weaknesses
1. The framework is only applicable to classification tasks. Extracting text feature space and projecting image features requires class labels from the dataset. Since the OT framework operates after feature extraction for label prediction, it cannot be adapted to non-classification tasks (e.g., LVLMs where CLIP vision encoders are used without text labels). It is a critical weakness as CLIP models are increasingly deployed in diverse scenarios without text labels.
2. Image features are projected using class-informed text features, meaning class labels influence feature extraction. This creates an uneven comparison baseline against methods extracting image features without label information.
3. Projecting image features onto the text subspace is theoretically most effective when adversarial perturbations lie in the orthogonal complement subspace. The paper also notes that adversarial perturbations distort image features along directions away from the text feature subspace in line 121. However, the paper lacks the quantitative analysis of perturbation reduction post-projection and the empirical validation of the orthogonality assumption

(Minor)

4. Implementation details (perturbation size, attack methods) for TeCoA/FARE baselines are omitted.
5. The δ₁ = δ₂ = δ equality in Appendix A.1 seems inadequate.
6. Figure labeling errors: Swapped descriptions between panels (a) and (b) in Figure 1's caption. Reversed "Original CLIP" and "Attacked CLIP" labels in Figure 1-(d)'s legend.

---

> ### Author Rebuttal · Authors · 2025-07-30
>
> **Q1:** Applicability beyond classification tasks.
> **A:** Our method is specifically designed to enhance the adversarial robustness of CLIP in classification settings, where the text labels (i.e., class names) are the predefined categories used to classify images at inference time.
> While our current framework is designed for classification tasks with a predefined class set, we acknowledge the importance of extending it to open-ended settings. A promising direction is to extract pseudo-labels from captions via clustering or noun phrase parsing, enabling the construction of a latent text feature space without explicit labels. This would allow projection and OT-based alignment in more general scenarios, which we consider a compelling direction for future work.
>
>
> **Q2:** Label influence on feature extraction fairness.
> **A:** We would like to clarify that CLIP-based classification is a closed-set problem, where all class names are predefined and available during inference. Our method projects image features into the subspace spanned by all class text embeddings, without using the ground-truth labels of test samples, thereby avoiding any label leakage. Moreover, the baselines we compare against (e.g., TTC, TeCoA) also rely on class text embeddings during inference, ensuring that all methods are evaluated under consistent and fair settings.
>
>
> **Q3:** Analysis of perturbation reduction and validation of orthogonality assumption.
> **A:** We thank the reviewer for raising this important point. We provide both theoretical justification and empirical validation:
> - **Perturbation reduction after projection.**
>   As shown in Equation (12) and formally proven in Appendix A.1, we demonstrate that projecting adversarial features onto the text subspace reduces semantic distortion:
>   $$
>   \Delta_\Pi \leq \Delta,
>   $$
>   where $\Delta$ and $\Delta_\Pi$ denote the cosine deviation before and after projection, respectively. This is further supported by Figure 4, which shows that projected features exhibit significantly higher similarity to text embeddings than the adversarial ones.
>
> - **Validation of the orthogonality assumption.**
>   We decompose adversarial perturbations into components parallel and orthogonal to the text subspace and find that the orthogonal component consistently dominates. For example, under PGD-1/255 on ImageNet, $\left\| \boldsymbol\delta_\perp \right\| / \left\| \boldsymbol\delta \right\| \approx 75.4\%,$
>   This confirms that most adversarial perturbations lie outside the semantic subspace.
>
>
>
> **Q4:** Implementation details for TeCoA and FARE.
> **A:**  We appreciate the reviewer’s suggestion. In our experiments, we did not re-implement TeCoA or FARE, but utilized the fine-tuned models released by TTC [51] to extract features. We have added the specific attack settings (including perturbation size, attack type, and evaluation protocol) used in the original TeCoA and FARE papers to the revised version.
>
>
> **Q5:** $\boldsymbol\delta_1 = \boldsymbol\delta_2 = \boldsymbol\delta$ assumption.
> **A:** The assumption $\boldsymbol\delta_1 = \boldsymbol\delta_2 = \boldsymbol\delta$ was made for simplicity, to emphasize the core insight: **projection suppresses cosine similarity distortion under adversarial perturbations**. We have extended the analysis in the appendix to cover the general case where $\boldsymbol\delta_1 \ne \boldsymbol\delta_2$. Specifically, under first-order approximation, the cosine distortion before projection becomes:
> $$\Delta \approx \left| \mathbf{x}_1^\top \boldsymbol{\delta}_2 + \mathbf{x}_2^\top \boldsymbol{\delta}_1 - s(\mathbf{x}_1^\top \boldsymbol{\delta}_1 + \mathbf{x}_2^\top \boldsymbol{\delta}_2) + \boldsymbol{\delta}_1^\top \boldsymbol{\delta}_2 \right| + O(\epsilon^2),
> $$
>
> and after projection:
>
> $$
> \Delta_\Pi \leq |s - s_\Pi| + \epsilon \left( \frac{1}{\|\mathbf{\Pi} \mathbf{x}_1\|} + \frac{1}{\|\mathbf{\Pi} \mathbf{x}_2\|} \right).
> $$
>
> This yields the bound:
> $$
> \frac{\Delta_\Pi}{\Delta} \lesssim \frac{1}{1 + |s|} < 1,
> $$
> The result confirms that projection consistently mitigates cosine similarity distortion even under mismatched perturbations, with $s = \cos(\mathbf{x}\_1, \mathbf{x}2)$ and $s\_\Pi = \cos(\mathbf{\Pi} \mathbf{x}\_1, \mathbf{\Pi} \mathbf{x}\_2)$.
>
> **Q6:**  Figure labeling errors.
> **A:** We thank the reviewer for pointing out the labeling errors. We have corrected them in the revised version.
>
> **Q7:** Why was ε=1/255?
>  **A:**
> We follow prior works such as TTC and TeCoA in using ε = 1/255 to ensure a fair and consistent comparison. We additionally conduct experiments under ε = 2/255 with ViT-B/16. Results are reported below and again confirm the superiority of our approach.
> | Model     | Attack Budget     | 9-datasets (Clean) | 9-datasets (Robust) | ImageNet (Clean) | ImageNet (Robust) |
> |-----------|-------------------|--------------------|----------------------|------------------|--------------------|
> | CLIP      | $\epsilon=2/255$  | 63.6               | 0.7                  | 67.6             | 0.2                |
> | TTC       | $\epsilon=2/255$  | 61.3               | 13.1                 | 67.1             | 19.6               |
> | **Ours**  | $\epsilon=2/255$  | **64.0**           | **19.5**             | **68.9**         | **31.2**           |

---

> > ### Comment · Reviewer_BF28 · 2025-08-06
> >
> > Thank you for your kind and detailed reply to my review. Most of my questions and concerns have been resolved, particularly regarding the orthogonality assumption, fairness in feature extraction, and the implementation details of the baseline methods. Your explanations have clarified these points. I also acknowledge that the proposed method demonstrates strong empirical performance and is well-supported by rigorous theoretical analysis.
> >
> > As I mentioned in my initial review, and as you also pointed out, the proposed approach is limited to classification tasks with pre-defined labels. In practice, the CLIP vision encoder is increasingly used not merely as a classifier, but as a visual embedding extractor integrated into complex systems. In such cases, where there are no pre-defined labels, applying the proposed method to obtain more robust embeddings remains challenging. I consider this a key limitation of the work.
> > While most of my concerns have been addressed, the limitation still remains.
> >
> > Therefore, I will raise my score to borderline accept.

---

> > > ### Author Response · Authors · 2025-08-06
> > >
> > > We’re glad our rebuttal has addressed your concerns and appreciate your recognition of the method’s empirical results and theoretical soundness. We agree that reliance on predefined labels is a key limitation and will make this clearer and discuss possible extensions in the final version. We sincerely thank you for raising your score.

---

### Official Review · Reviewer_JTxh · 2025-07-01

**Clarity:** 3
**Significance:** 3
**Originality:** 3
**Rating:** 5
**Confidence:** 3

**Summary:**

This paper proposes a training-free method to enhance the robustness of CLIP models against adversarial images. It introduces two alignment strategies: global feature alignment and local structural alignment, and employs an optimal transport-based framework to refine the correspondence between text and image embeddings. The method is evaluated on diverse datasets, and the results demonstrate that the proposed COLA approach significantly outperforms both state-of-the-art fine-tuned and training-free methods.

**Questions:**

1. In Figure 1(d), why does the Original CLIP have the lowest accuracy, while the Attacked CLIP shows the highest accuracy?
2. PGD and CW are no longer the most effective adversarial attack methods. Could the author provide results against state-of-the-art attacks (e.g., AutoAttack) to better demonstrate the method's performance?

**Ethical Concerns:**

["NO or VERY MINOR ethics concerns only"]

**Final Justification:**

The authors' rebuttal has successfully addressed my concerns. The proposed method is highly innovative and demonstrates strong performance across multiple datasets. I believe this research makes a meaningful contribution to improving the robustness of multimodal models. Therefore, I recommend accepting the paper.

**Limitations:**

yes

**Quality:**

3

**Strengths And Weaknesses:**

Strengths:
1. Theoretical analysis ensures the reliability of the proposed method.
2. The introduced global feature alignment and local structural alignment respectively target cross-modal and intra-modal feature alignment, offering valuable insights.
3. The experimental results are comprehensive and the performance is impressive.

Weaknesses:
1. The paper does not provide a clear framework diagram or pseudo-code, which makes it somewhat difficult for readers to fully understand the proposed method's workflow.
2. The evaluation is limited to CLIP-based classification tasks, without demonstrating whether the proposed approach can be extended to more complex models (e.g., vision-language models) or more challenging tasks (e.g., image captioning).

---

> ### Author Rebuttal · Authors · 2025-07-30
>
> **Q1:**   Framework diagram or pseudo-code.
> **A:** Thank you for the helpful suggestion. In the revised version, we have added a framework diagram in Section 3 to clearly illustrate the main components of our method, including subspace projection and OT-based alignment. We have also included pseudo-code in the appendix to describe the inference procedure in detail.
>
> **Q2:** Extending to more complex models or tasks.
> **A:** Our current work focuses on CLIP-based classification tasks, which rely on computing similarity scores between visual features and predefined textual class embeddings.  Tasks like image captioning involve language generation, which falls outside the design scope of our OT-based framework.
> We have conducted additional experiments using **SigLIP** and **EVA-CLIP**，two recent and stronger CLIP variants. As shown in the table below, our method consistently improves robustness under adversarial attacks, demonstrating its effectiveness across different vision-language model backbones.
> | Model        | Backbone  | 9-datasets (Clean) | 9-datasets (Robust) | ImageNet (Clean) | ImageNet (Robust) |
> |--------------|-----------|--------------------|----------------------|------------------|--------------------|
> | SigLIP         | SigLIP  | 65.6               | 1.5                  | 70.2             | 0.5                |
> | TTC          | SigLIP  | 62.3               | 15.8                 | 68.1             | 22.6               |
> | **Ours**     | SigLIP  | **65.0**           | **24.7**             | **70.9**         | **35.3**           |
> | EVA-CLIP           | EVA-CLIP  | 64.5              | 1.0                  | 68.8             | 0.4                |
> | TTC          | EVA-CLIP  | 61.2               | 13.7                 | 67.4             | 21.6               |
> | **Ours**     | EVA-CLIP  | **65.2**           | **18.9**             | **69.6**         | **33.7**           |
>
>
> **Q3:** Labeling in Figure1(d).
> **A:** We thank the reviewer for pointing this out. The legend in **Figure 1(d)** was mistakenly swapped between "Original CLIP" and "Attacked CLIP". We have corrected this labeling error in the revised version.
>
>
> **Q4:** Results of autoattacks.
> **A:** We have conducted additional experiments using **AutoAttack** on ViT-B/16. As shown in the table below, our method consistently improves robustness on both **9-datasets** and **ImageNet**, demonstrating effectiveness beyond specific attack types. We have included these results in the revision.
> | Model       | Attack Type   | 9-datasets (Clean) | 9-datasets (Robust) | ImageNet (Clean) | ImageNet (Robust) |
> |-------------|------------|--------------------|----------------------|------------------|--------------------|
> | CLIP        | autoattacks  | 63.6               | 0.5                  | 67.6             | 0.2                |
> | TTC         | autoattacks   | 61.3               | 8.3                 | 67.1             | 16.2               |
> | **Ours**    | autoattacks   | **64.0**           | **18.9**             | **68.9**         | **29.6**           |

---

> > ### Comment · Reviewer_JTxh · 2025-08-04
> >
> > Thanks to the authors for their efforts during the rebuttal process. My concerns have been addressed. I will maintain my score and recommend the acceptance of this paper.

---

> > > ### Author Response · Authors · 2025-08-04
> > >
> > > We sincerely thank you for your thoughtful comments and for your support of our work. We're glad that our rebuttal has addressed your concerns, and we truly appreciate your recommendation for acceptance.

---

### Official Review · Reviewer_i8r4 · 2025-07-03

**Clarity:** 3
**Significance:** 2
**Originality:** 2
**Rating:** 5
**Confidence:** 2

**Summary:**

## Summary
The authors propose a novel test-time counterattack for adversarial perturbations: their method is composed mainly of two steps,
 - a projection of the input image vectors (obtained by cropped, resized, and flipped views of the input image) into a "class subspace" obtained by the principal components of the class text prompts,
 - an optimal transport phase that for each class computes the optimal transport between the multiple image projections and the multiple class vectors specific for that class. The class with the minimum optimal transport plan is used as the prediction.

These two operations are both relatively inexpensive to adopt at inference time and seem to greatly increase the robustness of CLIP against adversarial attacks.

**Questions:**

# Questions

1. Are you the first to use optimal transport in such a way? If not, please provide a citation in the method section. If yes, then it would be better to properly mention it among your contributions.
In particular, in lines 142-144, you write *"$C_y^Π \in R^{N×M}$  denotes the transportation cost between the N augmented image views and the M textual descriptions of class $j$, which is usually quantified using the cosine similarity"*.
Who are you referring to with "usually quantified"?
2. Does the time reported in Table 8 include the computation of the PCA? How much time does optimal transport require?
3. The authors mention (and visualize) the misalignment gap between textual and image inputs in CLIP; however, it seems that there is no citation to works related to that issue.

**Ethical Concerns:**

["NO or VERY MINOR ethics concerns only"]

**Final Justification:**

Since the authors addressed my concerns in the rebuttal, I have decided to increase my score from "borderline accept" to "accept".
The issues I was concerned about were mainly regarding:
 - Lack of citation for certain statements in the article.
 - Relatively weak analysis in terms of runtime of different methods.

**Limitations:**

To be efficient COLA requires: (i) the computation of the multiple textual "views" of a class from an LLM, (ii) the estimation of the class PCA subspace. If the classes are static, these can be seen as inexpensive; however, if the textual prompts are highly dynamic, these may have a substantial impact on prediction time.
While this is an uncommon scenario, mentioning this limitation might be reasonable.

**Paper Formatting Concerns:**

The paper seems to be in line with the provided formatting instructions.

**Quality:**

2

**Strengths And Weaknesses:**

## Strengths

1. The paper is generally clearly written and readable.
2. The proposed approach is simple yet effective; in most of the results, it even outperforms the baseline CLIP model.
3. From the ablation studies and the theoretical analysis, it seems that both components (the projection and the optimal transport) improve adversarial robustness and even overall accuracy.

----

## Weaknesses

1. The authors mention multiple times that their proposed method is more efficient than alternatives; however, the computation time analysis is somewhat lacking and not as comprehensive as other tests. I would have appreciated more comparisons and experiments in that scenario.

2. While the proposed method is effective as a defence against attacks, it is still relatively expensive: it requires a class "augmentation" step with an LLM, estimation of the principal components, computation and projection of multiple input features, and finally, the optimal transport algorithm itself.  Admittedly, the first two steps can be cached (per class), but the projection and optimal transport must be computed for all inputs.

3. I am not sure if the advancements proposed by the paper are enough to garner a publication at this venue. However, being that I am not particularly familiar with this specific setting, I am conflicted regarding this point.

---

> ### Author Rebuttal · Authors · 2025-07-30
>
> **Q1:** Computation time analysis and comparisons.
> **A:** We have conducted additional experiments on running time, as shown in the table below. These results indicate that our method (COLA) not only requires less time compared to TTC but also achieves better performance.
> |Model| Dataset| Running Time | Clean Accuracy (%) | Robust Accuracy (%) |
> |--------------|-------------|--------------|---------------------|----------------------|
> | CLIP         | ImageNet   | 10min        | 62.1               | 1.1                  |
> | TTC          | ImageNet  | 40 min       | 51.7                | 40.0                 |
> | **COLA (Ours)** | ImageNet  | 28 min (Projection: 4 min, OT:8 min)      | 62.8             | 50.0                 |
> | CLIP         | Caltech101   | 2min        | 85.7               | 14.7                  |
> | TTC          | Caltech101  | 5 min       | 86.5                | 65.8                 |
> | **COLA (Ours)** | Caltech101  | 3 min (OT: 1min)      | 88.1             | 75.3                 |
> | CLIP         | DTD  | 0.5 min       | 40.6               | 3.0                  |
> | TTC          | DTD  | 3 min       | 37.0               | 27.3                 |
> | **COLA (Ours)** | DTD  | 1 min (OT: 0.5 min)       | 41.0               | 34.0
>
>
> **Q2:** Computational cost for projection and optimal transport.
> **A:** The projection step involves simple matrix multiplication between a $d$-dimensional input feature and the class-specific projection matrix, with a per-input cost of $\mathcal{O}(d^2)$, which is negligible in practice (e.g., <0.01s per sample). For OT, its complexity is generally $\mathcal{O}\left(n^2 \cdot \log\left(\frac{1}{\varepsilon}\right)\right)$, where $\varepsilon$ is the regularization coefficient and usually set to 0.01. Importantly, the OT computation is performed in parallel across all classes, resulting in efficient runtime. As reported in the table in response to Q1, the time costs are 8 minutes on ImageNet, 1 minute on Caltech101, and 0.5 minute on DTD.
>
> **Q3:** Explanation for OT.
> **A:** To the best of our knowledge, we are the first to apply OT for improving the adversarial robustness of CLIP. Prior works have used OT in other tasks such as distribution calibration [17] and few-shot learning [22].
> Regarding the cost metric, cosine similarity is commonly used in multimodal OT-based alignment. Prior works such as PLOT [8], AWT [56], and ALIGN [44] adopt it to measure semantic distances between visual and textual representations.
> We have revised the manuscript accordingly.
>
> **Q4:** Details in Table 8.
> **A:** The time reported in Table 8 includes the PCA computation. OT computation takes approximately 8 minutes on ImageNet, 1 minute on Caltech101, and 0.5 minute on DTD, as shown in the table provided in response to Q1.
>
> **Q5:** Citation for CLIP modality misalignment.
> **A:** We have revised Section 1 and Section 3.2 to include relevant citations. Li et al. [25] and Mao et al. [30] demonstrate that adversarial perturbations and low-level visual biases can disrupt the alignment between visual and textual features. More recent studies, such as [r1] and [r2], further characterize the modality gap in CLIP.
> [r1] Stabilizing  Modality Gap & Lowering Gradient Norms Improve Zero-Shot Adversarial Robustness of VLMs, KDD 2025
> [r2] Mitigate the Gap: Improving Cross-Modal Alignment in CLIP, ICLR2025
>
> **Q6:** Impact of dynamic prompt generation.
> **A:** In CLIP-based classification, class names are fixed, and prompts are generated using simple templates. For example, CLIP uses “a photo of a [class name],” which is shared across all test samples. Thus, prompt generation incurs only a one-time cost per class and does not affect per-image inference time. While dynamic prompts per image could introduce overhead, such cases are rare and fall outside the scope of CLIP-based classification. We have included this limitation in the revision.

---

### Decision · Program_Chairs · 2025-09-17

**Decision:**

Accept (spotlight)

**Comment:**

Contribution: it proposes a training-free method to enhance the robustness of CLIP models against adversarial images with two alignment strategies: global feature alignment and local structural alignment. It also employs an optimal transport-based framework to refine the correspondence between text and image embeddings. The method is evaluated on diverse datasets, and the results demonstrate that the proposed method significantly outperforms both state-of-the-art fine-tuned and training-free methods.

Ratings: the final scores are Accept (x2) and Borderline Accept(x2). Overall, the reviewers find the method to be technically solid, theoretically well-motivated, and empirically effective. So, it is a clear Accept.